# Quantifying methane emissions from Queensland's coal seam gas producing Surat Basin using inventory data and a regional Bayesian inversion

Ashok K. Luhar[1], David M. Etheridge[1], Zoë M. Loh[1], Julie Noonan[1], Darren Spencer[1], Lisa Smith[2], and Cindy Ong[3]

[1]CSIRO Oceans and Atmosphere, Aspendale, Victoria 3195, Australia
[2]Katestone Environmental Pty. Ltd., Milton, QLD 4064, Australia
[3]CSIRO Energy, Kensington, WA 6152, Australia

*Correspondence to*: Ashok Luhar (Ashok.Luhar@csiro.au)

**Abstract.** Methane ($CH_4$) is a potent greenhouse gas and a key precursor of tropospheric ozone, itself a powerful greenhouse gas and air pollutant. Methane emissions across Queensland's Surat Basin, Australia, result from a mix of activities, including the production and processing of coal seam gas (CSG). We measured methane concentrations over 1.5 years from two monitoring stations established 80 km apart on either side of the main CSG belt located within a study area of $350 \times 350$ $km^2$. Using an inverse modelling approach coupled with a bottom-up inventory, we quantify methane emissions from this area. The inventory suggests that the total emission is $173.2 \times 10^6$ kg $CH_4$ $yr^{-1}$, with grazing cattle contributing about half of that, cattle feedlots ~ 25%, and CSG Processing ~ 8%. Using the inventory emissions in a forward regional transport model indicates that the above sources are significant contributors to methane at both monitors. However, the model underestimates approximately the highest 15% of the observed methane concentrations, suggesting underestimated or missing emissions. An efficient regional Bayesian inverse model is developed, incorporating an hourly source-receptor relationship based on a backward-in-time configuration of the forward regional transport model, a posterior sampling scheme, and the hourly methane observations and a derived methane background. The inferred emissions obtained from one of the inverse model setups that uses a Gaussian prior whose averages are identical to the gridded bottom-up inventory emissions across the domain with an uncertainty of 3% of the averages best describes the observed methane. Having only two stations is not adequate at sampling distant source areas of the study domain, and this necessitates a small prior uncertainty. This inverse setup yields a total emission of $165.8 \times 10^6$ kg $CH_4$ $yr^{-1}$, slightly smaller than the inventory total. However, in a subdomain covering the CSG development areas, the inferred emissions are $63.6 \times 10^6$ kg $CH_4$ $yr^{-1}$, 33% larger than those from the inventory. We also infer seasonal variation of methane emissions and examine its correlation with climatological rainfall in the area.

# 1 Introduction

Methane ($CH_4$) is a potent greenhouse gas with a global warming potential 84 times greater than carbon dioxide ($CO_2$) over a 20-year period and 28 times greater over a 100-year period (IPCC, 2014). It is emitted by both anthropogenic activities (e.g. coal mining and the raising of cattle) and natural sources (e.g. wetlands). In terms of anthropogenic radiative forcing, methane is the second most important greenhouse gas after $CO_2$. Globally averaged surface $CH_4$ concentrations have increased by almost 160% since pre-industrial times, from 731 ppb (by volume) in 1750 to 1859 ppb in 2018 (Meinshausen et al., 2017;

WMO, 2018; Rubino et al., 2019), and this increase has been largely due to changes in anthropogenic methane (IPCC, 2014). Compared to $CO_2$, the atmospheric lifetime of methane is much shorter (~ 10 years), which means that near-term warming of the climate could diminish following mitigation actions that reduce methane emissions. Being chemically reactive, methane also plays an important role as a precursor to tropospheric ozone, itself a greenhouse gas and an air pollutant affecting human health and plant productivity. Thus, understanding and quantifying methane emissions at various scales is crucial to studying

changes in atmospheric radiative forcing and air quality.

Globally, a top-down estimate over the period 2008-2017 suggests that agriculture and waste contribute to about 60% of the total anthropogenic methane emissions, followed by fossil fuel production and use (gas, oil, coal mining and industry) at 31% (Saunois et al., 2020). However, a study using measurements of carbon-14 in methane recently showed that nearly all methane from fossil sources is anthropogenic, contrasting with the bottom-up estimates of significant natural geologic seepage (Etiope

et al., 2019; Etiope and Schwietze, 2019), and that fossil fuel methane emissions may be underestimated by up to 40% (Hmiel et al., 2020). Significant $CH_4$ emissions from conventional and unconventional gas fields have been reported in the scientific literature (e.g., Brandt et al., 2014; Schneising et al., 2014; Alvarez et al., 2018).

In the Australian state of Queensland, since the mid-2000s there has been a rapid growth of the production of coal seam gas (CSG), which is virtually pure methane (Towler et al., 2016; DNRM, 2017). CSG, also known as coalbed methane, is classed

as an unconventional natural gas, typically extracted from coal seams at depths of 200–1000 m. As of 2015-16, 96% of the gas production in Queensland was CSG, with most of it coming from the Surat Basin (78%, 21187 $Mm^3$) and the rest (18%, 4958 $Mm^3$) from the Bowen Basin (DNRM, 2017). With the sharp rise of CSG production, methane emissions from the Surat Basin are a focus of Australia's CSIRO Gas Industry Social and Environmental Research Alliance (GISERA) (https://gisera.csiro.au) research in Air Quality and Greenhouse Gas. The Surat Basin is predominantly rural, and methane sources other than CSG

include agriculture and coal mining. CSG activities that lead to potential methane emissions include CSG wells, pumps, pipelines, vents, pneumatic controls, and produced water bodies (Day et al., 2013).

The objective of the present paper is to quantify methane emissions from a region of $350 \times 350$ $km^2$ of Queensland's side of the Surat Basin (Figure 1, covering the area 148° 17' 43.4"–151° 49' 30.5" E, 25° 3' 48.8"–28° 5' 3.7" S) that encompasses the main CSG production and processing areas using a top-down approach coupled with a bottom-up emission inventory that

serves as a prior. The latter involves deriving emissions through a compilation of sources and activity data and application of

emission factors. We conducted concurrent in-situ atmospheric monitoring of methane during July 2015 – December 2016 at two locations, namely Ironbark and Burncluith, 80 km from each other. The two stations were setup such that they were on either side of the broad present and projected CSG work area in the Surat Basin. The measured concentrations allow for an atmospherically based validation of the bottom-up inventory by using it in a forward mesoscale meteorological and transport model, namely TAPM (see Section 4.1), and comparing the predicted methane concentrations with the measurements at the two sites.

An efficient top-down, or inverse, modelling methodology for regional scale (~ 100–1000 km) is formulated and applied to quantify $CH_4$ emissions in the Surat Basin. It combines a Bayesian inference approach, an hourly-averaged high-resolution backward-in-time construction of the forward model TAPM, and a posterior probability density function (PDF) sampling scheme. A method to correct for time-lag effects in the backward plume methodology is presented. The 1.5 years long hourly methane measurements from the two stations are combined in a Bayesian calculation to derive a top-down emission distribution. Methane background calculation and filtering methodologies are devised. Various Bayesian priors and their uncertainties, including the use of the bottom-up emissions to act as a prior, are tested. The inferred top-down $CH_4$ emissions are examined alongside the bottom-up inventory emissions for the whole study domain as well as subdomains containing the CSG and non-CSG activities. We also compare the performance of the top-down method by comparing the modelled methane concentrations obtained using the top-down derived emissions in forward modelling with the observed concentrations. To our knowledge, this study is the first in Australia to quantify regional scale $CH_4$ emissions through a top-down approach employing transport modelling and concentration measurements, although studies at other spatial scales with broadly similar approaches have been reported, e.g. by Luhar et al. (2014) and Feitz et al. (2018) for single point sources at local scale and by Wang and Bentley (2002) at continental scale with Australian methane emissions divided into eight source regions.

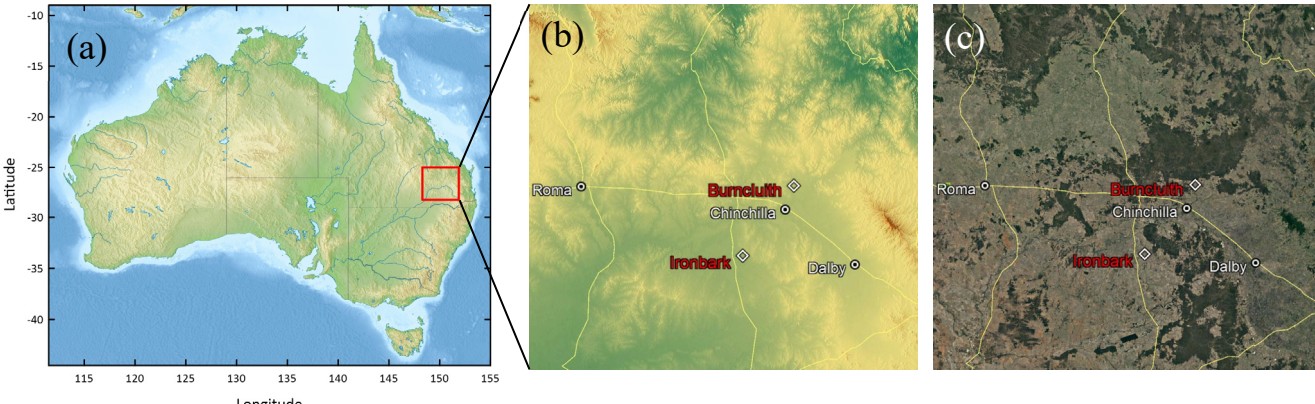

**Figure 1.** (a) Map of Australia showing the 350 × 350 km² study domain (red square) of Queensland's part of the Surat Basin. The base relief map is from **https://www.mapsland.com/oceania/australia/large-relief-map-of-australia** (used under Creative Commons Attribution-ShareAlike 3.0 Licence); (b) orography of the study domain, with terrain elevation ranging approximately between 100 m (green) and 1140 m (red) above sea level; (c) a Google Earth map of the study domain showing the surface characteristics. The Ironbark and Burncluith monitoring sites, and the three biggest towns of Dalby, Roma and Chinchilla (population ~ 12700, 6850 and 6600, respectively) in the area are also shown in (b) and (c).

## 2 Monitoring and data filtering

We set up two monitoring stations, namely Ironbark (150° 14' 37.6" E, 27° 8' 6.6" S; 226.806 km east, 6995.596 km north MGA (Map Grid Australia), Zone 56) and Burncluith (150° 42' 5.4" E, 26° 34' 2.4" S; 271.051 km east, 7059.430 km north MGA, Zone 56), located about 80 km apart on two sides of the main coal seam gas belt of the Surat Basin (Figure 1b,c). The selection of the site locations was largely based on a meteorological and dispersion modelling study (Day et al., 2015; Etheridge et al., 2016) that suggested that with the prevailing winds from the north-east and south-west quadrants, long-term continuous monitoring of greenhouse gas concentrations at these two locations would optimise the size and frequency of detection of methane emissions from the broader CSG source region without being unduly impacted by individual sources in the proximity of the measurement sites. There were other practical considerations, namely access, power, security, landowner assistance and possible future developments that would impact the site.

Continuous high frequency (~ 0.3 Hz) measurements of the concentrations of $CH_4$, $CO_2$ and water vapour (and also carbon monoxide (CO) at Burncluith) were made at the two sites for about three years with an overlapping period of 1.5 years (July 2015 to December 2016) using Picarro cavity ring down spectrometers (model G2301 at Ironbark, and G2401 at Burncluith) with inlets placed on masts at a height of 10 m. The installations are described by Etheridge et al. (2016). Measured concentrations (strictly speaking, mole fractions in dry air, also volumetric mixing ratios) from each site can be exactly intercompared due to identical calibrations and measurement methodologies. The additional CO measurements at Burncluith are useful in detecting combustion sources of $CO_2$ and $CH_4$. Measurement accuracy was better than ± 0.1 ppm for $CO_2$ and ±

1 ppb for $CH_4$ (Etheridge et al., 2014). Concurrent meteorological observations included winds measured at 5.8 m above ground level (AGL) at Ironbark and at 7.6 m AGL at Burncluith using sonic anemometers.

The Burncluith station was located on a private farm and there were 30–40 cattle in the adjoining paddocks. Occasionally, under suitable meteorological conditions with the cattle upwind of the inlet, the emissions from the local cattle caused one or many sharp peaks in the observed methane signal, typical of a nearby point source. We developed a method which removes these sharp, transient peaks but does not alter the underlying signals from the numerous, region-wide feedlots, grazing cattle, or other sources. This filtering method is described in the Supplement S1.1 and, for consistency, was also applied to the data from Ironbark, although local cattle are less in number and further away at this site.

Frequently, high methane concentrations at the two sites were observed at night under light wind stable conditions, particularly at Burncluith. Despite being of much practical interest, however, light winds are difficult to represent in a mesoscale meteorological and transport model. The causes for that include inadequate physical understanding of light-wind processes, flow properties being very sensitive to local topography, and model resolution constraints (Luhar and Hurley, 2012). As a practical measure, we filtered out the nighttime sampling hours for light wind conditions, and this method is described in the Supplement S1.2.

Methane emissions due to biomass burning are not part of the bottom-up inventory that we consider in the present modelling due to their being sporadic and highly unpredictable. Enhanced levels of $CH_4$ and CO were detected at Burncluith during forest fires in the northern sector of Burncluith and wood-heater operations from the property located in the proximity of the monitoring station. The observed CO was used to filter out these occasional biomass burning events from the measured concentration time series, an approach similar to that used by Jeong et al. (2012). Details of the CO filter are given in the Supplement S1.3.

The number of data hours after the filtering was 6432 for Ironbark and 4149 for Burncluith (cf. the original, valid number of data points of 10938 and 12660, respectively). Unless stated otherwise, the filtered $CH_4$ data were used for our analysis and modelling.

## 3 Bottom-up emission inventory

Activity data for the year 2015 were used to develop a bottom-up emission inventory for methane for the Surat Basin. The emission inventory covered a domain of $345 \times 345$ km$^2$ with a spatial resolution of $1 \times 1$ km$^2$. Standard methodologies were generally adopted with data from various State and Federal Government Departments (e.g. (National Pollutant Inventory (NPI), National Greenhouse and Energy Reporting (NGER), and National Resource Management (NRM)). The bottom-up inventory included the following fourteen emission sectors: (1) feedlots, (2) grazing cattle, (3) piggeries, (4) poultry farms, (5) power stations, (6) coal mining, (7) CSG processing, (8) CSG production, (9) domestic woodheating, (10) vehicular traffic, (11) land-fills, (12) sewage treatment plants, (13) river seepage, and (14) geological seepage. The first four can be grouped as agricultural

activities. The inventory excluded $CH_4$ emissions from burning of biomass, land clearing, termites, ground-water wells (that were registered), wetlands, or fuel consumption and any material handling related to mining activities. Additional details pertaining to the bottom-up inventory compilation are briefly given in the Supplement S2, with a full report (Katestone, 2018) given in the Supplement S6.

Figure 2 presents the bottom-up inventory emissions attributed to the various sectors in the Surat Basin, with the total emissions being $173.2 \times 10^6$ kg $CH_4$ yr$^{-1}$. Grazing cattle has the largest contribution, followed by cattle feedlots and CSG processing. We use this emission inventory for our study duration, July 2015–December 2016, with the assumption that any emission changes from the year 2015 to 2016 were insignificant. It is also assumed that all emissions are invariant with time. Although diurnal and seasonal variations for some emissions, viz. wood-heating, traffic, and power plant, are available in the raw data used in the inventory, contributions from these emissions are amongst the smallest and, therefore, we averaged these emissions over the full year for the purpose of computational efficiency in the modelling conducted here.

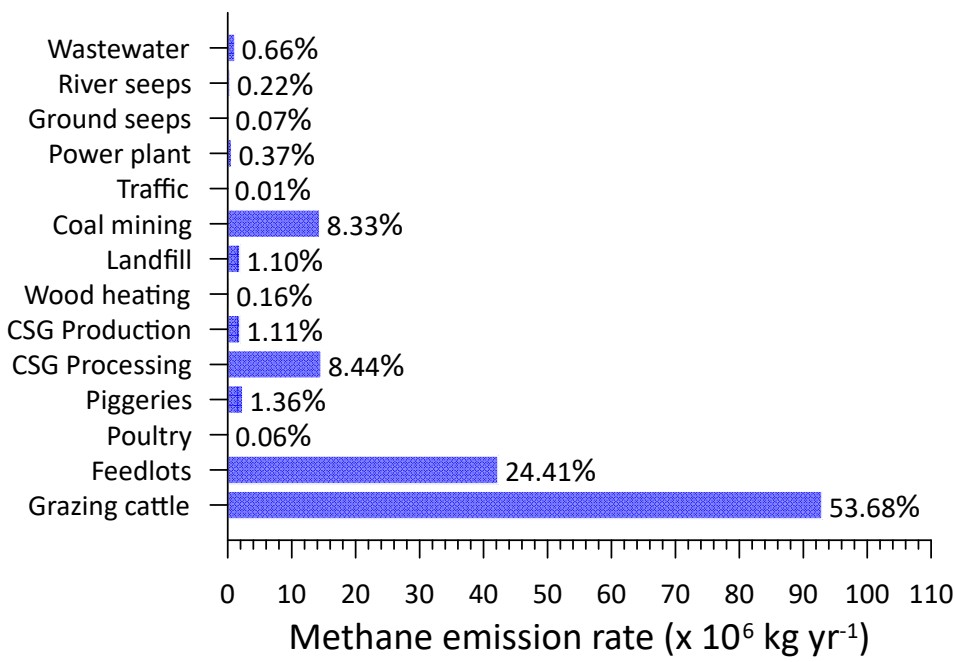

**Figure 2. Bottom-up methane inventory emissions from the Surat Basin by sector/source; % of the total also shown. The total emission is $173.2 \times 10^6$ kg $CH_4$ yr$^{-1}$.**

Figure 3a presents the distribution of inventory methane emissions (kg yr$^{-1}$ gridcell$^{-1}$) regridded at a resolution of $5 \times 5$ km$^2$ ($69 \times 69$ grid points). There are localised sources as well as extensive, uniformly distributed source areas. The latter are emissions due to grazing cattle. These emissions are plotted in Figure 3b in which four different coloured areas are the so-

called National Resource Management (NRM) regions. In each of these regions the available total number of grazing cattle was distributed uniformly, with the total number of grazing cattle in the study area being 1,086,059. There were 235 cattle feedlots and Figure 3c shows the distribution of their emissions. These are localised, but distributed throughout the region, with some located between the two monitoring stations. Two mining source areas are also located between the two monitoring stations (Figure 3d).

The CSG emissions are shown in Figure 3e (processing) and Figure 3f (production). The CSG production emissions are from wellhead (separators, wellhead control equipment, maintenance and leaks), combustion (flaring, well head pumps, backup generators, and diesel used by vehicles) and pipeline emissions (high point vents on produced water pipelines and pipeline control equipment) (Day et al., 2013). The CSG processing sources consist of processing facility emissions (control equipment, compressor venting, and gas conditioning units), combustion emissions (flaring, plant compressors, backup generators, and diesel used by vehicles), and collection and storage of water produced. Emissions from some of the CSG sources are continuous while others are intermittent (however, the inventory assumes all CSG emissions are time invariant). There were 5 CSG operators with 13 processing facilities and 4628 wells within the study domain. The well numbers included CSG producing (~ 85%) as well as exploration/appraisal/capped wells. Because of insufficient information, methane emissions from two of the five operators are not part of the inventory, but it was established that these two operators, with a total of 256 wells, only accounted for about 1.5% of the CSG activities that may be related to emissions. The biggest contributor to the total CSG methane emissions was venting (88%) from processing. Methane from produced water is a component of both CSG production and processing is an important source (Iverach et al., 2015). It was included under venting and was calculated at $1.63 \times 10^6$ kg yr$^{-1}$ (~10% of the total CSG emissions). Contribution from flaring was about 8%.

All major bottom-up emissions, namely from grazing cattle, feedlots, CSG processing and production, and coal mining, have potentially significant uncertainty, arising from uncertainty in both the activity data and emission factors, for example their potential temporal variation and how up to date they are with respect to the study period considered.

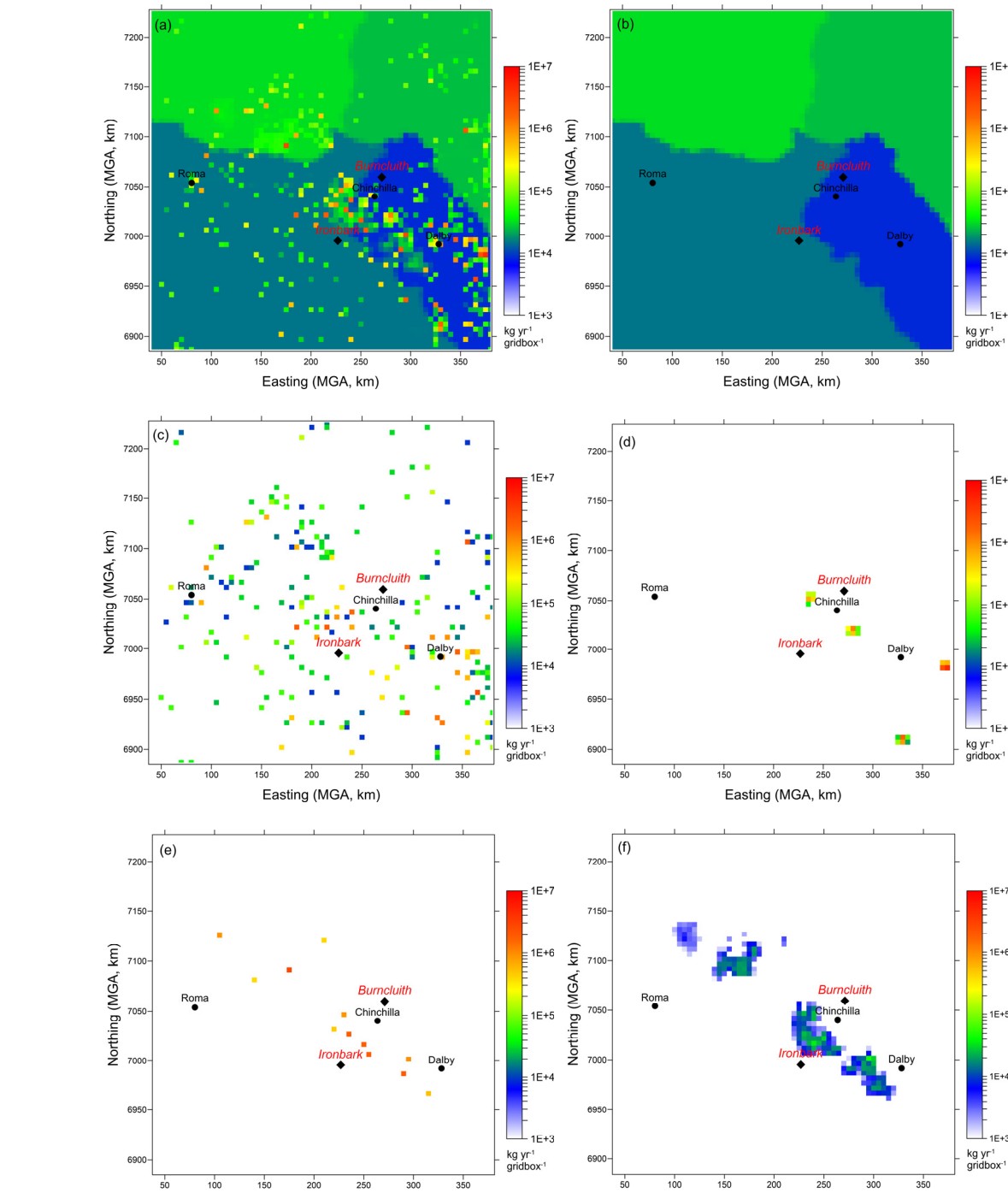

**Figure 3. Bottom-up methane inventory emissions from the Surat Basin (kg CH₄ yr⁻¹ gridbox⁻¹, the grid-box size is 5 × 5 km²). Also shown are the Ironbark and Burncluith monitoring sites, and the three biggest towns. (a) All emissions, and those due to (b) grazing cattle, (c) cattle feedlots, (d) coal mining, (e) CSG processing, and (f) CSG production.**

## 4 Modelling regional methane using the bottom-up emission inventory

We use the above inventory emissions in a (forward) regional meteorological and transport model and compare the modelled methane with the ambient measurements from the two sites.

### 4.1 Model and configuration

The prognostic mesoscale model used is The Air Pollution Model (TAPM vn4.0.5) developed by CSIRO, which has coupled meteorological and dispersion components and is designed for applications ranging in scale from local to regional ($\sim$ < 1000 km) (Hurley et al., 2005; Hurley, 2008).

The meteorological component of TAPM predicts the local-scale flow against a background of larger-scale meteorology provided by the input synoptic-scale analyses (or forecasts). It solves momentum equations for horizontal wind components;
the incompressible continuity equation for the vertical velocity in a terrain-following coordinate system; and scalar equations for potential virtual temperature, specific humidity of water vapour, cloud water/ice, rainwater and snow. Explicit cloud microphysical processes are included. Pressure is determined from the sum of hydrostatic and optional non-hydrostatic components, and a Poisson equation is solved for the non-hydrostatic component (not used here). Turbulence closure in the mean prognostic equations uses a gradient diffusion approach with non-local or counter-gradient corrections, which depends
on eddy diffusivity ($K$) and gradients of mean variables and a mass-flux approach. The eddy diffusivity $K$ is determined using prognostic equations for the turbulent kinetic energy ($E$) and its dissipation rate ($\varepsilon$). A vegetative canopy, soil scheme, and urban scheme are used at the surface, while radiative fluxes, both at the surface and at upper levels, are also included. Surface boundary conditions for the turbulent fluxes are determined using the Monin-Obukhov similarity theory and parameterisations for stomatal resistance.

The dispersion module makes use of the predicted finer-scale meteorology and turbulence fields from the meteorological component and comprises a default Eulerian grid-based conservation equation for species concentration (Hurley et al., 2005). The model has previously been applied to a variety of flow, turbulence and dispersion problems at various scales, such as those reported by Luhar and Hurley (2003), Luhar et al. (2008), Hurley and Luhar (2009), Luhar and Hurley (2012), Luhar et al. (2014), Matthaios et al. (2017), and Luhar et al. (2020), which include model evaluation studies.

TAPM can be used in a one-way nestable mode to improve efficiency and resolution. The global databases input to the model include land use, terrain height, leaf-area index, synoptic-scale meteorological reanalyses, and sea-surface temperature.

We applied TAPM for the duration 1 July 2015 – 31 December 2016 with two nested domains for both meteorology and dispersion: $370 \times 370$ km$^2$ with grid resolution $5 \times 5$ km$^2$ and $1110 \times 1110$ km$^2$ with grid resolution $15 \times 15$ km$^2$. Both domains had $75 \times 75$ grid points and were centred on (150°4.5' E, 26°35' S), which is equivalent to 208.657 km east and 7056.383 km
north MGA. There were 25 vertical levels, of which the lowest four were 10 m, 25 m, 50 m and 100 m AGL. The input synoptic-scale fields of the horizontal wind components, temperature and moisture required as boundary conditions for the

outermost model domain were sourced from the U.S. NCEP (National Centers for Environmental Prediction) reanalysis database given at a resolution of 2.5° latitude × 2.5° longitude at 6-hourly intervals (Kalnay et al., 1996; https://psl.noaa.gov/data/gridded/data.ncep.reanalysis.html). The model outputs hourly-averaged fields of meteorology and concentration.

The bottom-up inventory emissions lie within the inner model domain. In this model setup, each inventory emission grid cell (at $5 \times 5$ km$^2$) was considered as a surface source, apart from the emissions from the power stations which were taken as point sources with specification of their stack heights and plume-rise parameters. For computational efficiency, rather than considering all 14 emission categories plotted in Figure 2 as separate sources, we aggregated them into 9 sectors with each sector taken as a tracer source: Grazing cattle (Source 1); Feedlots, Piggeries and Poultry (Source 2); CSG Processing (Source 3); CSG Production (Source 4); Mining (Source 5); River seeps (Source 6); Domestic wood heating, Wastewater treatment and Motor vehicles (Source 7); Ground seeps and Landfill (Source 8); and Power stations (Source 9). The relative emissions (%) of the above nine Sources are 53.8, 25.8, 8.4, 1.1, 8.3, 0.21, 0.82, 1.2 and 0.37%.

## 4.2 Estimation of background methane concentration

Since the simulated methane does not include the background levels that are representative of methane emissions located outside the bottom-up inventory, we devised a method for estimating hourly varying background CH$_4$ for each site involving concentrations under high atmospheric mixing conditions and the hourly standard deviation of concentration (see details in the Supplement S3). The estimated background concentration can be either added to the simulated methane or subtracted from the observed methane.

The estimated background methane concentration time series for Ironbark and Burncluith look very similar, and in Figure 4 we present the average (green line) of the two background time series. The plot shows a marked seasonal variation in the background methane with a peak in September (early spring) and a minimum in February (late summer). To view the background variation with respect to the measured methane signal, we also present in Figure 4 as dot points the unfiltered hourly mean observations (clipped at 2100 ppb) at Ironbark. The uncertainty (one standard deviation) in the background CH$_4$ is 3.6 ppb and 3.3 ppb for Ironbark and Burncluith, respectively. The difference between the estimated background at Ironbark and that at Burncluith (purple line in Figure 4) is small and within ± 5 ppb. Any difference between the two backgrounds could be due to different sites in the study area getting impacted by different out-of-domain emissions depending on the transport meteorology. On average, the background concentration at Ironbark is greater by 1 ppb, and the standard deviation of the difference is 1.4 ppb. The average of the two background time series is taken to represent the regional hourly background CH$_4$ concentration, with an average uncertainty of 3.5 ppb. Sensitivity of the inferred emissions to other choices of the background concentration is examined in Section 7.4.

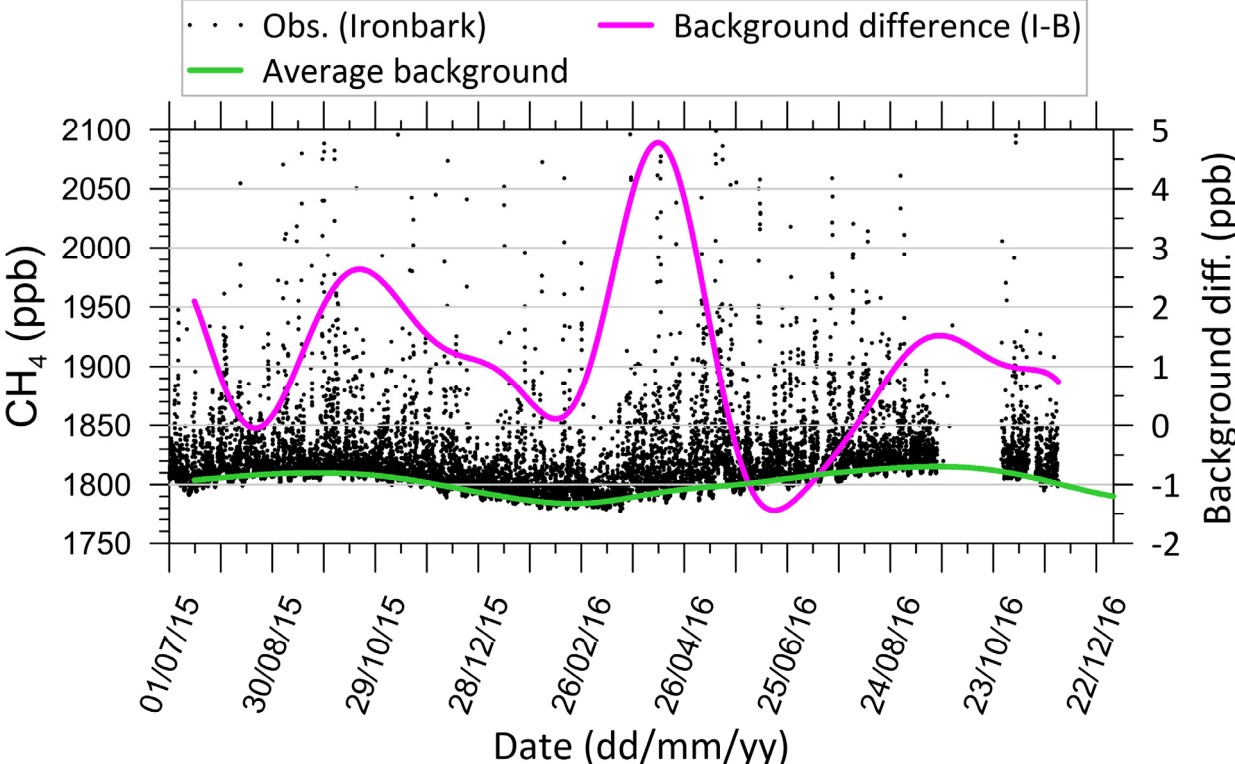

**Figure 4. Estimated average hourly-averaged background CH₄ concentration time series (green line), and the difference between the estimated backgrounds between Ironbark and Burncluith (purple line). The data points are the hourly mean measurements at Ironbark without any filtering (clipped at 2100 ppb to make the background concentration variation stand out better).**

### 4.3 Model performance for meteorology

Accurate modelling of the flow field over our region of interest is important as it controls the atmospheric plume transport and dispersion which in turn influences the accuracy of prediction of CH₄, and conversely the accuracy of inferred emissions. The hourly-averaged predicted winds extracted from the model output for the inner nest at the lowest model vertical level (10 m) at the grid point nearest to each of the two monitoring stations were compared with the observations from the two stations for the duration of the simulation. The details of the model performance for meteorology is given in the Supplement S4. At both sites, the measured winds were most frequent from the north-east sector, with those at Burncluith being generally weaker in strength than those at Ironbark. As judged from the correlation coefficient ($r$) and index of agreement (IOA) values, the performance of TAPM for wind speed and wind direction was comparable to that obtained in other TAPM modelling studies (see the Supplement S4).

## 4.4 Modelled methane compared to observations

The monitoring sites were selected to avoid potential large, sustained methane sources within 10-20 km or even small sources within about a kilometre of the measurement inlet. Small sources that were closer to the inlets (mainly Burncluith) were identified and their signals filtered from the data as described in Section 2. As a result, we expect that the hourly-averaged filtered data are as representative as possible of the atmospheric methane concentration across the 5 × 5 km model grid cell containing the observation site, and can be directly compared to the model simulations.

The hourly-averaged modelled methane concentrations on the innermost grid domain were extracted at the lowest model level at the grid point nearest to each of the monitoring sites for comparison with the observations. The hourly-averaged concentrations simulated for the individual 9 source categories were aggregated and added to the estimated background concentration to compare with the observed, filtered $CH_4$ concentrations.

The scatter plots in Figure 5 comparing the modelled and observed $CH_4$ at the two sites display a substantial degree of scatter, which is not unusual for atmospheric transport and diffusion models driven by predicted meteorology and using hourly-averaged concentrations paired in both time and space (e.g. Luhar et al., 2008). While the correlation coefficient values of 0.57 and 0.74 for Ironbark and Burncluith, respectively, imply a reasonable model prediction (see Table 1 for additional model performance statistics for the inventory emissions), it is clear that the modelled levels are generally lower than the observations, particularly the higher-end concentrations at Ironbark.

There could be various reasons for the differences between the modelled and observed methane, including uncertainty associated with the bottom-up emission inventory, its potential temporal variation, sources missing from the emission inventory, potential changes to the 2015 bottom-up inventory used here in the year 2016 (see Section 7.4), and the general modelling uncertainty, including that related to representing point measurements by grid-cell averaged model values.

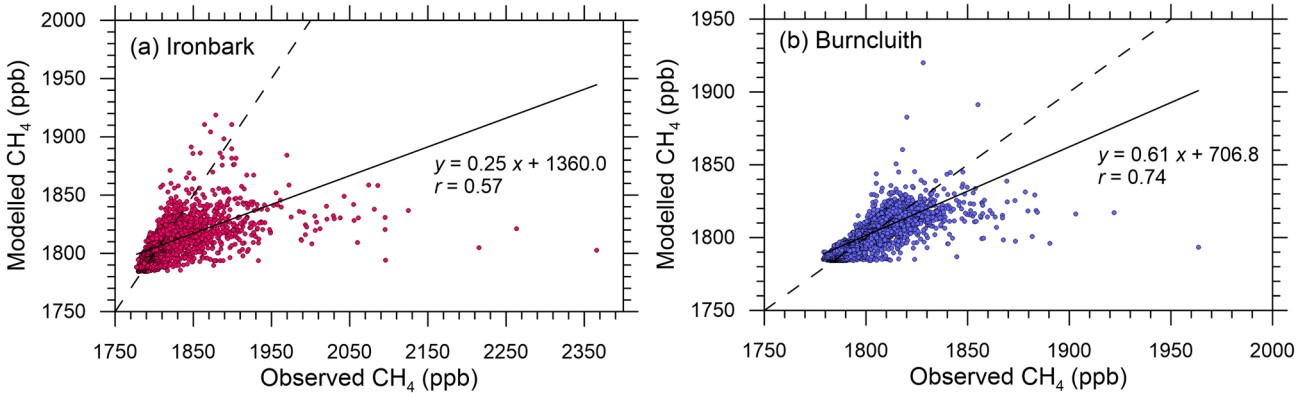

**Figure 5. Hourly-averaged observed methane plotted against the simulated methane for the two monitoring stations. The solid line is the least-squares fit, and the dashed line is the 1:1 line (i.e. perfect agreement).**

The comparison in Figure 5 involving hourly methane paired in time and space enables a simple, yet stringent, validation check of a transport model, especially one that is driven by turbulent flow fields predicted by a prognostic meteorological model instead of observations. A complementary but less stringent approach in validating air quality models is the quantile-quantile (q-q) plot, which is a graphical technique for testing "goodness of fit" between two distributions. In such a plot, typically, sorted modelled concentrations are plotted against sorted observed values (i.e. unpaired in time) at a monitoring

location (e.g., Venkatram et al., 2001; Luhar and Hurley, 2003; http://www.itl.nist.gov/div898/handbook/eda/section3/qqplot.htm). If the two sets come from a population with the same distribution, the data points should fall approximately along the 1:1 line. The principal advantage of a q-q plot is that a "good fit" is easy to recognize, and various distributional aspects, such as shape, tail behaviour and outliers, can be simultaneously examined.

In the q-q plot in Figure 6 for Ironbark, the observed $CH_4$ distribution is modelled well for measurements < 1820 ppb, but for higher observed concentrations, which account for approximately 25% of the sample size, the modelled values are smaller. For Burncluith, the q-q plot shows a substantially better model performance, with the model underestimation of higher-end (> 1820 ppb) methane observations, which is approximately 10% of the sample size, much reduced compared to Ironbark. Overall, TAPM is largely predicting the observed $CH_4$ distribution correctly, except for a relatively few higher-end concentrations.

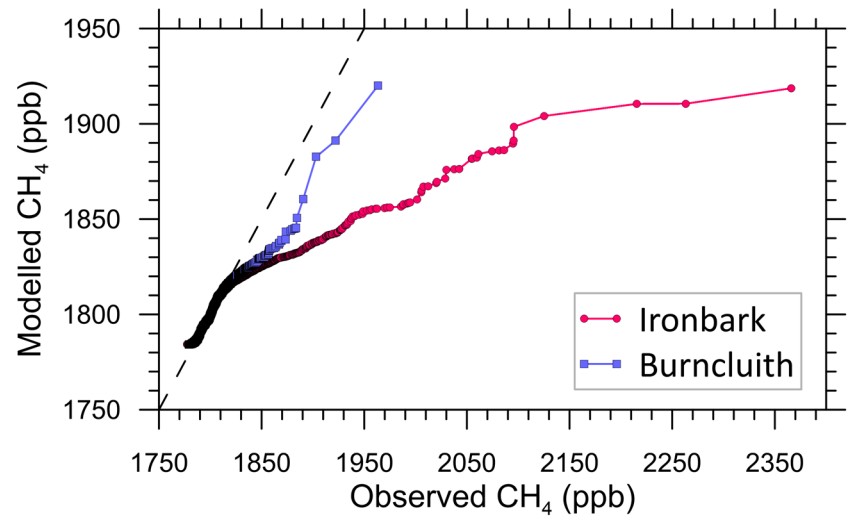

**Figure 6. Q-q plot showing the sorted hourly-averaged observed $CH_4$ concentrations versus the sorted modelled ones at Ironbark and Burncluith. The line of perfect agreement (dashed line) is also shown.**

**4.5 Contribution to the modelled methane by various source categories**

The top four source categories based on their contribution to the modelled $CH_4$ averaged over the full study period at Ironbark were Source 1 (45%, Grazing cattle), Source 2 (25%, Feedlots, Piggeries and Poultry), Source 3 (19%, CSG Processing), and Source 5 (5.5%, Mining). These were the same at Burncluith, but with their respective contributions being 69%, 17%, 6.4% and 4.1%. The CSG Production (Source 4) contributions are 2.2% and 0.73%, respectively, at the two sites.

In contrast, the largest four contributors to the highest 5% of the modelled hourly-averaged methane concentrations (i.e. all the concentrations above the 95th percentile) at Ironbark turn out to be Source 3 (35%), Source 2 (27%), Source 1 (25%) and Source 5 (7%). These at Burncluith are Source 1 (28%), Source 2 (25%), Source 3 (22%) and Source 5 (13%). The CSG Production (Source 4) contributes 3.8% and 2.5%, respectively, at the two sites. The Source 2 grouping is dominated by Feedlots.

The CSG Processing (Source 3) emissions are localised near the two sites which result in methane spikes under favourable winds and thus contribute more to the higher-end modelled methane than to the overall average methane. In contrast, the simulation average methane is dominated by Sources 1 and 2 because concentration enhancements due to these sources occur under most wind conditions because of their very wide distribution across the region.

**5 Regional top-down, or inverse, modelling for emission estimation**

Given that the bottom-up emission inventory underestimates the observed methane in the Surat Basin, then one may ask what is the magnitude and distribution of methane emissions that is implied by the methane concentration measurements at Ironbark and Burncluith? This is addressed by the inverse modelling approach for regional emissions formulated and applied below.

**5.1 Bayesian inverse modelling approach**

Our inverse model uses a Bayesian inference approach that incorporates, a source-receptor relationship, concentration measurements, and prior information on source parameters (i.e. source information obtained independently of the measurements) (Rao, 2007; Singh et al., 2015). The approach updates the source prior as concentration measurements are considered, and accounts for both model and observational uncertainties.

Several applications using the Bayesian approach have previously been conducted for methane source estimation, including those at local scale (Yee and Flesch, 2010; Luhar et al., 2014; Feitz et al., 2018) and regional scale (Jeong et al., 2012; Miller et al., 2014; Henne et al., 2016; Cui et al., 2017).

The approach hinges on Bayes' theorem (Jaynes, 2003):

$$p(\mathbf{q}|\mathbf{c}) = \frac{p(\mathbf{c}|\mathbf{q}) \cdot p(\mathbf{q})}{p(\mathbf{c})}, \tag{1}$$

where the *prior* PDF $p(\mathbf{q})$ reflects our knowledge of the source parameter vector $\mathbf{q}$ prior to receiving the concentration observations $\mathbf{c}$; $p(\mathbf{c}|\mathbf{q})$ is the *likelihood* function which is the probability of experiencing $\mathbf{c}$ for a given $\mathbf{q}$ and is typically

obtained using a model-derived source-receptor linkage; the *posterior* $p(\mathbf{q}|\mathbf{c})$ relates to the update of $p(\mathbf{q})$ by its modulation by $p(\mathbf{c}|\mathbf{q})$ which contains the new information brought in by the concentration measurements $\mathbf{c}$; and $p(\mathbf{c})$ [=$\int p(\mathbf{c}|\mathbf{q})p(\mathbf{q})d\mathbf{q}$] is the *evidence* and is basically a normalisation constant in the present application (Yee and Flesch, 2010). The likelihood function, also termed the source-receptor relationship, is derived using a transport and dispersion model.

It is assumed that the number of sources ($N_s$) and their locations $(\mathbf{x}_{s,1}, \dots, \mathbf{x}_{s,j}, \dots, \mathbf{x}_{s,N_S})$ where $\mathbf{x}_{s,1} \equiv (x_{s,1}, y_{s,1}, z_{s,1})$ are given

*a priori* and the source emissions are positive and non-zero. The emission rates of these sources are to be estimated, and these are represented by $\mathbf{q} \equiv (q_1, \dots, q_j, \dots, q_{N_S})$ with a total of $N_S$ unknown emission rates. Assuming each source emission to be independent, the prior PDF can be written as:

$$p(\mathbf{q}) = \prod_{j=1}^{N_S} p(q_j). \tag{2}$$

Assuming that the model and measurement uncertainties are independent and distributed normally, the total likelihood of all $\mathbf{c}$ for a given hypothesis of $\mathbf{q}$ is calculated as (Yee, 2012)

$$p(\mathbf{c}|\mathbf{q}) = \prod_{i=1}^{N_m} \frac{1}{\sqrt{2\pi}\left(\sigma_i^2 + \sigma_{m,i}^2\right)^{1/2}} \exp\left\{-\frac{\left(c_{m,i}(\mathbf{q}) - c_i\right)^2}{2\left(\sigma_i^2 + \sigma_{m,i}^2\right)}\right\}, \tag{3}$$

$\mathbf{c} \equiv (c_1, \dots, c_i, \dots, c_{N_m})$, $c_i$ is the observed concentration at $i$-th instant (time and location), $c_{m,i}$ is the corresponding modelled concentration for a given hypothesis of $\mathbf{q}$, $\sigma_i$ is the independent measurement error, $\sigma_{m,i}$ is the independent model error, $N_m$ is the number of concentration data (which can be time series from several independent monitors). $c_{m,i}$ for all hypotheses, or possible values, for $\mathbf{q}$ is calculated and used in constructing the likelihood distribution $p(\mathbf{c}|\mathbf{q})$. Hence the posterior PDF for a given source hypothesis $\mathbf{q}$ is calculated as:

$$p(\mathbf{q}|\mathbf{c}) = \frac{1}{Z_0} \prod_{j=1}^{N_S} p(q_j) \prod_{i=1}^{N_m} \frac{1}{\sqrt{2\pi}\left(\sigma_i^2 + \sigma_{m,i}^2\right)^{1/2}} \exp\left\{-\frac{\left(c_{m,i}(\mathbf{q}) - c_i\right)^2}{2\left(\sigma_i^2 + \sigma_{m,i}^2\right)}\right\}, \tag{4}$$

where $Z_0$ is equivalent to $p(\mathbf{c})$ and is essentially a normalisation constant. The posterior yields probabilities of all emission rates ($\mathbf{q}$) considered.

The total modelled concentration at a given location $\mathbf{x}_r$ and time is determined as

$$c_{m,i} = \sum_{j=1}^{N_s} c_{m,ij}. \tag{5}$$

Because methane is treated as a passive tracer, the concentration field simulated for one rate of emission can be scaled linearly for another without the need to re-run the model. Thus

$$c_{m,ij} = q_j \alpha_{ij}(\mathbf{x}_{s,j}, \mathbf{x}_{r,i}), \tag{6}$$

for each emission rate component of $\mathbf{q}$. The quantity $\alpha_{ij}(\mathbf{x}_{s,j}, \mathbf{x}_{r,i})$ is the source-receptor relationship or coupling coefficient and is equivalent to the modelled mean concentration at a given time and location $\mathbf{x}_{r,i}$ due to $j$-th source release at location $\mathbf{x}_{s,j}$ with a unit emission rate.

In Eq. (4), in the absence of an informative prior, a uniform prior PDF can be used with the given limits ($q_{max}$, $q_{min}$)

$$p(q_j) = \frac{1}{q_{max,j} - q_{min,j}}, \tag{7}$$

with the probability being zero outside these bounds.

If the prior is Gaussian, then

$$p(q_j) = \frac{1}{\sqrt{2\pi}\,\sigma_{p,j}} \exp\left\{-\frac{(q_j - q_{p,j})^2}{2\sigma_{p,j}^2}\right\}, \tag{8}$$

where $q_p$ and $\sigma_p$ are the prior mean emission rate and its standard deviation, respectively.

High dimensionality of the posterior makes its direct computation and the subsequent integration (the 'brute-force' method) over the source-parameter space very expensive or perhaps even impossible. For Gaussian priors and uncertainties, the posterior can be solved for the mean and variance with their analytical matrix forms (Tarantola, 2005; Jeong et al., 2012). To make the inverse approach more generally applicable and efficient, we use a Markov chain Monte Carlo (MCMC) technique incorporating the Metropolis-Hastings algorithm to sample the posterior PDF (Tarantola, 2005; Yee, 2012). With MCMC, non-Gaussian priors or uncertainties, or parameters with known physical constraints can also be included (Miller et al., 2014). The normalization constant $Z_0$ in Eq. (4) need not be known before MCMC samples can be drawn from the posterior PDF. This ability to generate a sample without knowing this constant of proportionality (which is often extremely difficult to compute) is a major feature of MCMC algorithms (Luhar et al., 2014). The frequency distribution of the MCMC-generated samples represents the posterior.

The posterior PDF can be marginalized to obtain the mean emissions rate for each source as follows:

$$\bar{q}_j = \int q_j\, p(\mathbf{q}|\mathbf{c})\, d\mathbf{q}, \tag{9}$$

and likewise, the variance can also be determined.

## 5.2 Construction of the hourly source-receptor relationship

In order to use hourly measurements, the source-receptor relationship needs to be calculated every hour for every source (real or potential) location and every monitor location using either forward or backward transport modelling (Rao, 2007). Generally speaking, if the number of source locations under consideration is greater than the number of receptor locations (as for the present case) then the backward approach is much more computationally efficient (Luhar et al., 2014).

In the backward approach, source emissions are tracked backwards in time from a monitor treated as a source. The value at a
given point of the constructed backward concentration field is analogous to the magnitude of contribution made by an emitting source at that point to the true (i.e. forward) modelled concentration at the monitor. Hence, we can use a single backward source-receptor relationship distribution determined every hour to get the contribution made by each real or potential source located in the domain. This contrasts with the forward modelling approach in which each source location must be considered as a unique, separate source and its dispersion computed for every hour. Essentially, the source-receptor relationship furnishes
a way to chart the distribution of source potential within given geographical domain. However, it does not quantitatively allocate the real contribution of sources within the domain to the concentration levels detected at monitoring stations— this is done by the Bayesian inference (Eq. (4)).

One backward approach for regional scale is to use backward trajectories constructed by only using three-dimensional winds computed from a meteorological model (e.g., Cheng et al., 1993). However, such wind trajectories only represent advective
transport and do not account for turbulent mixing which causes a plume to disperse as it travels in the atmosphere. If measurements given at a high temporal resolution, e.g. hourly averages, are to be used for inversion it is necessary that the influence of atmospheric flow and dispersion processes that occur at such scales is considered. This can only be properly done by simulating backward tracer plumes which considers both advection and turbulent mixing.

We modify TAPM to construct backward dispersing plumes. The Eulerian dispersion module in TAPM comprises a solution
of the advection-diffusion equation for the ensemble mean concentration $c$, which for a passive species is (e.g. Yee et al., 2008):

$$\frac{\partial c}{\partial t} + \overline{\mathbf{u}}.\nabla c - \nabla.\left(\mathbf{K}\,\nabla c\right) = S, \tag{10}$$

in which the unknown turbulent flux terms are closed using the $K$-theory or gradient transport approach. The forcing term $S$ represents species emissions. The elements of the eddy diffusivity tensor $\mathbf{K}$ are zero except along its main diagonal ($K_x$, $K_y$, $K_z$). Diffusion is assumed to be symmetric in the horizontal plane, so $K_x = K_y = K_H$ (say). $K_H$ and $K_z$ are determined using the
modelled turbulent kinetic energy (TKE) and the TKE dissipation rate.

The vertical component $\overline{w}$ of the mean wind vector $\overline{\mathbf{u}}$ ($\equiv \overline{u}, \overline{v}, \overline{w}$) in Eq. (10) is determined by using the continuity equation after the mean horizontal wind velocity components ($\overline{u}, \overline{v}$) are calculated.

The Eulerian adjoint of Eq. (10) describes the backward evolution of a scalar field ($c^*$), and is also termed backward or retro plume, adjoint function, sensitivity function, or influence function, and is given as (Marchuk, 1995; Pudykiewicz, 1998; Hourdin and Talagrand, 2006; Yee et al., 2008)

$$-\frac{\partial c^*}{\partial t} - \overline{\mathbf{u}}.\nabla c^* - \nabla.(\mathbf{K}\nabla c^*) = M, \tag{11}$$

where $M$ is the forcing term representing the measurement distribution, which is treated as a source at the measurement (or receptor) location. Therefore, $\alpha_{ij}$ in Eq. (6) is equivalent to $c^*$ derived for a unit emission rate.

The implementation of Eq. (11) in TAPM was done through changes in the forward model code as follows. The meteorological and turbulence fields calculated by the model at every hour (not hourly-averaged) were stored for the full simulation period. The modelled horizontal components ($\overline{u}, \overline{v}$) of wind were reversed (i.e. by sign change). The (inverted) vertical wind component ($\overline{w}$) was then calculated by solving the continuity equation given the reversed horizontal wind components. The turbulence parameter values remained the same. The diffusivities in the dispersion component are positive and do not have any correction for counter-gradient flux in the vertical, and, therefore, they were not modified for the backward mode. The two monitor locations were treated as separate 'sources' each having unit emission, and hourly-averaged plume dispersion fields due to these 'sources' was determined by running the TAPM dispersion module backwards in time for the entire simulation duration by using the reversed winds calculated previously. The meteorological and turbulence fields were linearly interpolated in time for dispersion calculations for model time steps lying between two successive hours. The resulting hourly-averaged backward concentration fields were used as the source-receptor relationship. For inversion, we assume that all methane sources are located near the ground within the lowest model level (i.e. 10 m AGL) and, therefore, only the 10-m hourly source-receptor relationship was required.

One complexity with doing a backward dispersion calculation using one continuous release over the full simulation period over a large domain, as done here, is that the source-receptor field at a given hour is a superposition of plume footprints from the current hour as well as previous hours (typically 4–5 hours for the present domain size). So, there is a time history of the plume in the source-receptor field at a given time (whose influence becomes smaller and smaller as the distance between the source and the receptor becomes smaller, the domain size decreases, the averaging time is increased, or when the winds are strong). However, this time history in a backward run corresponds to future hours in a forward run, so at a given hour there can be a time mismatch between the forward concentration at a grid point and the backward concentration at that point. One way to deal with this problem is to do a separate backward run for each hour for the whole simulation period; however, this is extremely expensive computationally. As a practical and approximate solution to this issue, at a particular backward travel hour ($t$) the plume travel time ($t_r$) from the release point (i.e. the monitor location) to a grid point ($\mathbf{x}$) is determined by releasing

a second tracer (with concentration $c^* = c_2^*$) backwards from the monitor simultaneously with the main tracer (with concentration $c^* = c_1^*$) with the same tracer properties except that it decays exponentially with a decay rate of $\lambda$ (taken as $10^{-6}$ s$^{-1}$), so

$$c_2^*(\mathbf{x}, t) = c_1^*(\mathbf{x}, t) \exp(-\lambda t_r), \tag{12}$$

which gives

$$t_r(\mathbf{x}, t) = \frac{1}{\lambda} \ln\left[\frac{c_1^*(\mathbf{x}, t)}{c_2^*(\mathbf{x}, t)}\right]. \tag{13}$$

The source-receptor value ($c^* = c_1^*$) calculated at a grid point location $\mathbf{x}$ at a given backward travel hour $t = t_b$ is then taken equal to that calculated at the same location at $t = t_b + t_r$ (where $t_r$ rounded to the nearest hour). The forward travel hour for a grid point is equal to the total hours in a simulation period minus $t_b$. Therefore, the source-receptor relationship ($c^*$) for the grid points at time $t$ is constructed from the output of $c_1^*$ at different times according to the value of $t_r$ at individual grid points. A maximum value for $t_r$ needs to be specified, which we take 15 h – approximately the time taken by the backward plume

from either monitor to leave the (innermost) model domain (beyond this value, $c^*$ is zero). This is needed to avoid occasional spurious smearing in the spatial patterns of $c^*$ caused by a very diluted, turning, or recirculating backward plume that has travelled longer than $t_r$ overlapping the direct backward plume at a particular location.

To illustrate the modelled forward and backward relationship and the impact of accounting for $t_r$, Figure 7a presents the hourly-averaged forward modelled 10-m concentration field ($c$) in the innermost model domain on 20 June 2016 at 2300 h (local

standard time) due to a sample of 12 point sources, all emitting at the same fixed rate and whose locations correspond to some of the feedlots. Figure 7b is the backward modelled 10-m concentration field ($c^*$) for Ironbark (I) at the same time without the travel time correction (i.e. $t_r = 0$), and Figure 7c is the same field with the travel time correction. Essentially, the value at any point in the backward field is equivalent to the forward model concentration value at Ironbark if there were a source at that point with the same emission rate (as the backward emission rate). The backward concentration value at a given location

represents the probability (including both frequency and intensity) a source emission at that location adds to the concentration at the monitoring site. The backward field is mainly determined by flow the field across the domain and the separation between the receptor and the source. Figure 7a suggests that only one source, S1, contributes to concentration at Ironbark. Figure 7c is consistent with this, in which the backward plume from Ironbark only impacts S1 with the same magnitude, and not any other source location. On the other hand, the backward plume in Figure 7b does not pass through any of the 12 sources, meaning no

impact of these sources at Ironbark, which obviously is not correct as S1 does impact Ironbark (Figure 7a). Figure 7c is the source-receptor relationship (normalised by the fixed emission rate) for Ironbark for the hour under consideration.

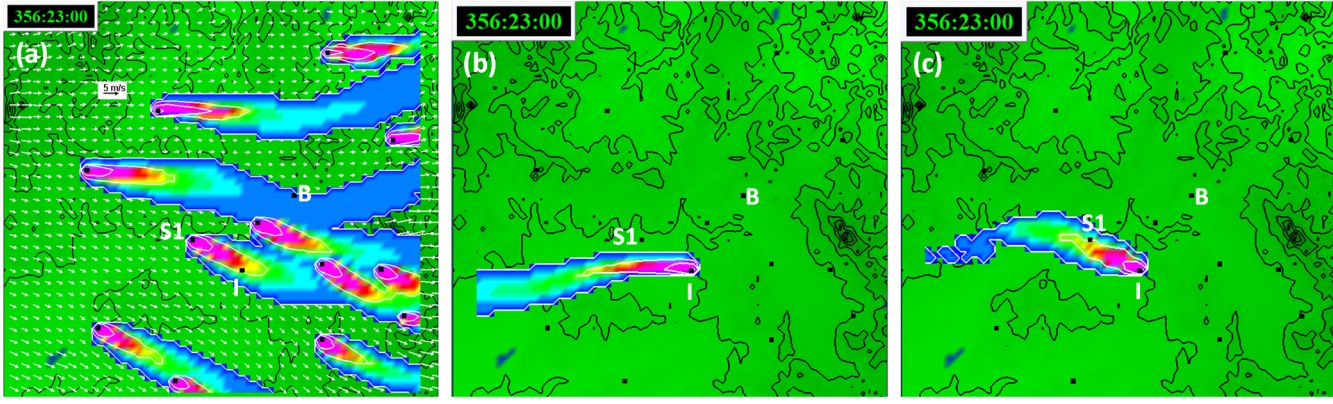

**Figure 7. (a)** Forward modelled hourly-averaged 10-m concentration field on 20 June 2016 at 2300 h (local standard time) due to 12 point sources, with the 10-m modelled winds also shown; **(b)** backward modelled 10-m concentration field for Ironbark (I) at the same time without the travel time correction ($t_r = 0$); and **(c)** backward modelled 10-m concentration field for Ironbark with the travel time correction. Each source point has the same emission rate. The plume contours (white) and colours represent the same concentration values. The black contours represent the topography. The model domain size is $370 \times 370$ km$^2$, and the Ironbark (I) and Burncluith (B) locations are shown.

A hourly-averaged modelled backward concentration field ($c^*/q$, s m$^{-3}$) at the lowest model level (i.e. 10 m AGL), an example of which was shown in Figure 7c, obtained for a unit emission rate ($q = 1$ g s$^{-1}$) is in essence the required hourly source-receptor relationship which can be linearly scaled for any other emission rate ($q$).

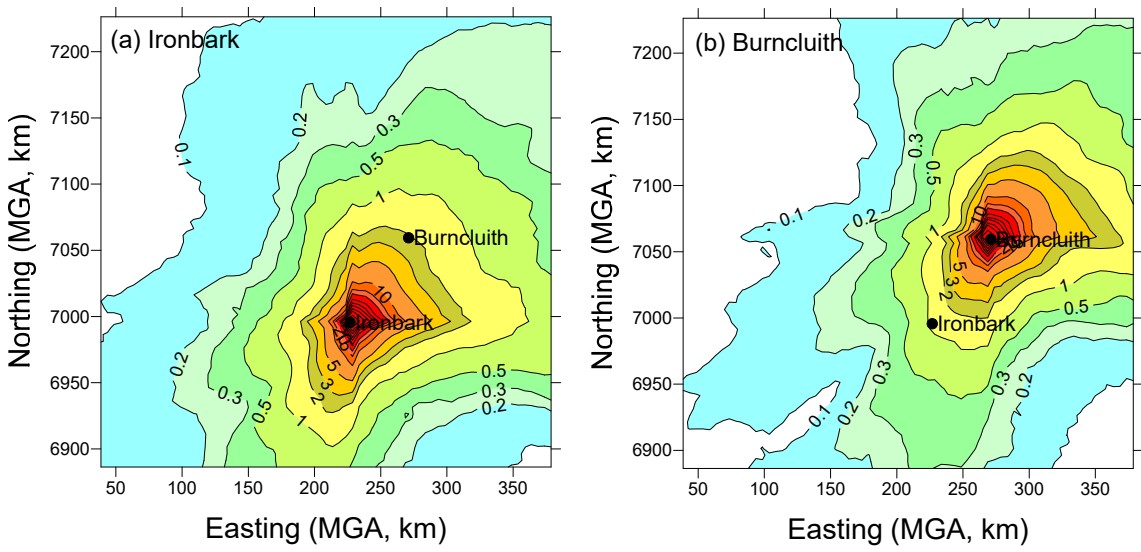

**Figure 8. Normalised modelled backward distribution of near-surface concentration ($c^*/q$, $\times\ 10^{-9}$ s m$^{-3}$), which is an average over the entire study period: (a) Ironbark, and (b) Burncluith.**

The modelled backward concentration field ($c^*/q$, s m$^{-3}$) averaged over all hourly fields over the simulation period (i.e. 1.5 years) for Ironbark is shown in Figure 8a, which suggests that, overall, any sources located farther from the monitoring station would contribute less as plume concentrations decrease with increasing distances, and vice versa. The directional distribution of the backward field is also a function of the distribution of regional winds which determine how often the receptor is downwind of a source (see wind roses in Figure S3). The values in the south-east and north-west corners of the study domain are particularly low, so potential sources located there would, on average, have relatively low probability of being sampled at Ironbark.

The backward distribution for Burncluith (Figure 8b) is very similar, but since it is located north of Ironbark it would sample potential sources in the north-east better.

The two monitoring sites combined would sample the bulk of the CSG sources between and around them in the domain (which was the prime objective of our monitoring).

**5.3 Bayesian inversion setup**

Assuming that emission rates are time invariant, we use all hourly methane data ($N_m$) from the two monitoring stations together in one combined Bayesian calculation to determine the total emission rates from gridded sources using Eq. (4). Since each hour corresponds to a unique meteorological condition, the use of all hours simultaneously provides the meteorological variability needed to achieve a better "triangulation" for source estimation. This approach is similar to that used by Luhar et al. (2014) in the context of a local point source. It requires the source-receptor matrix ($c^*(\mathbf{x}, t)$) for each hour for each measurement site (e.g. Figure 7c).

For the purposes of inferring emissions using our Bayesian methodology, the source array of 69 × 69 used in the forward modelling above is rather too large a source number to explore all the source possibilities (i.e. hypotheses) on hourly basis, even with use of the MCMC sampling. Moreover, there is only a limited amount of information available from just two monitoring sites. A coarser array of sources is more practicable, and consequently we consider an array of 11 × 11 localised sources ($N_s = 121$, cell size ~ 31 × 31 km$^2$) within the same model domain. No sub-grid variability of these emission sources is considered. The hourly source-receptor relationships calculated at 5 × 5 km$^2$ resolution for Ironbark and Burncluith were used. Our inverse methodology as used does not distinguish between different source categories. This is mainly because the concentration of methane alone was monitored and not tracers specific to methane source types. Therefore, there are no separate sources categories in the inferred emissions (unlike what was done for the forward simulation) and only total emissions are optimised.

To reduce serial correlations in the sequence of MCMC samples drawn from the posterior using the Metropolis-Hastings algorithm, we only retained every 5$^{th}$ sample. The total number of useable samples was 21,000 for each source, of which the first 1,000 samples were discarded as "burn-in" samples. The selected samples were then used in the calculation of the source statistics.

## 6 Inversion using 'synthetic' methane concentration data

A 'synthetic' inverse run is first performed by using the modelled hourly-averaged time series of methane concentration at Ironbark and Burncluith obtained using the bottom-up inventory (regridded to 11 × 11 sources, see Figure 9a, to be consistent with the source number considered in the inversion) to investigate whether the inverse methodology is able to retrieve the bottom-up emissions and under what type of priors and their uncertainties. The results of this exercise provide a useful guidance to the subsequent inversion using the actual measured methane data.

Only the forward modelled (or synthetic) concentrations at the two monitoring sites were used at times when valid (or filtered) methane observations were available ($N_m$ = 10581). The background measurement uncertainty was taken as $\sigma$ = 3.5 ppb based on the previous calculation, and the uncertainty in the transport model was assumed to be $\sigma_m$ = 20% of the modelled concentration (Yee and Flesch, 2010; Luhar et al., 2014). These values are also used later in Section 7 for inversions based on the methane data.

### 6.1 Selection of the prior

Specifying the prior PDF $p(\mathbf{q})$ is an important step, even for the present synthetic case because we are still limited to the same degree of information available (i.e. concentrations from only two sites), the same number of unknown sources to estimate, and the same domain size as in the inversion case with the real concentration data considered subsequently. We specify the following two Gaussian priors:

- An identical (or uniform) Gaussian $p(\mathbf{q})$ for each source with a mean methane emission rate $q_p$ = 45.4 g s$^{-1}$ (= 1.43 × 10$^6$ kg yr$^{-1}$) per source is specified, with a specified standard deviation $\sigma_p$. This mean value is essentially the total bottom-up emission from the domain divided by the number of sources (i.e. 121).
- The bottom-up inventory emissions as a Gaussian prior. The inventory emissions shown in Figure 9a are taken as the mean values of a Gaussian prior for each source, with a specified standard deviation $\sigma_p$.

### 6.2 Results for the synthetic case

In Figure 10a, the methane emission rates inferred by the inverse methodology for the uniform Gaussian prior case with a prior uncertainty of $\sigma_p$ = 5% of the mean for each source are plotted against the bottom-up inventory sources used to construct the synthetic concentration time series for the inversion. Ideally, the data points should fall along the 1:1 line, but due to the limited

amount of information supplied via the modelled concentrations from only two monitors and the prior being narrow and not very informative, most inferred emission rates are scattered around the prior mean, i.e. $q_p = 45.4$ g s$^{-1}$, although it is apparent that a few inferred emission rates are greater than this value and tending to the corresponding bottom-up emission rates. The spatial distribution of the inferred emissions is presented in Figure 9b, which, as expected, is much more uniform than the bottom-up inventory emissions in Figure 9a.

When the prior uncertainty is increased to $\sigma_p = 10\%$ of the mean (Figure 10b), the scatter increases, but most inferred emissions still stay around the prior mean, barring some higher-end ones which move further closer to the corresponding bottom-up emission rates. Further increase in $\sigma_p$ leads to a larger increase in scatter, with no improvement in the inferred emissions.

The total inferred methane emissions are $179.3 \times 10^6$ and $175.7 \times 10^6$ kg yr$^{-1}$ for $\sigma_p = 5\%$ and 10% of the mean, respectively – values very similar to the bottom-up inventory total of $173.2 \times 10^6$ kg yr$^{-1}$.

Figure 11a with 5% prior uncertainty is the same as Figure 10a except that the individual bottom-up inventory emissions (Figure 9a) have been used as the mean values of the Gaussian prior. The inversion retrieves the bottom-up emissions very well with a little scatter in the data points. The spatial distribution of the inferred emissions is presented in Figure 9c for this case, which is very similar to that of the inventory emissions in Figure 9a. As the prior uncertainty is increased to $\sigma_p = 10\%$ of the mean (Figure 11b), the uncertainty in the retrieved emissions gets larger, with a slight decrease in the correlation.

The total inferred emissions corresponding to Figure 11a and Figure 11b are $164.8 \times 10^6$ and $156.9 \times 10^6$ kg yr$^{-1}$, respectively – values somewhat smaller than the inventory total $173.2 \times 10^6$ kg yr$^{-1}$.

A comparison of Figure 9c with the bottom-up inventory (Figure 9a) indicates that some regions in the south-east, for example the strong coal mining source on the eastern boundary at the grid location (11, 4), and north west corners are not replicated by the inverse model. This is despite a strong prior with a relatively small uncertainty, and could be due to the fact that the two monitoring locations do not sample this source area sufficiently (see Figure 8) (because they were sited to optimally sample the CSG region). Extra monitoring stations and/or separate narrower priors for such sources would be needed to cover these areas better.

The synthetic case results suggest that with only two monitoring locations the bottom-up inventory Gaussian prior works well and is, indeed, needed. Obviously, a small prior uncertainty biases the inferred emission distribution towards the prior $p(\mathbf{q})$, and what uncertainty level is selected depends on the available information supplied to the inversion. The synthetic case reveals that $\sigma_p \sim 5\%$ of the mean is needed to retrieve the bottom-up emissions. Thus, for a real inversion using the methane measurements one may expect that an even narrower prior uncertainty may be needed. Further guidance on $\sigma_p$ can also come from a comparison of the forward modelled methane concentrations using the inferred emissions with the methane observations from the two sites.

The synthetic case results also demonstrated that the regional inverse model was stable and feasible with MCMC.

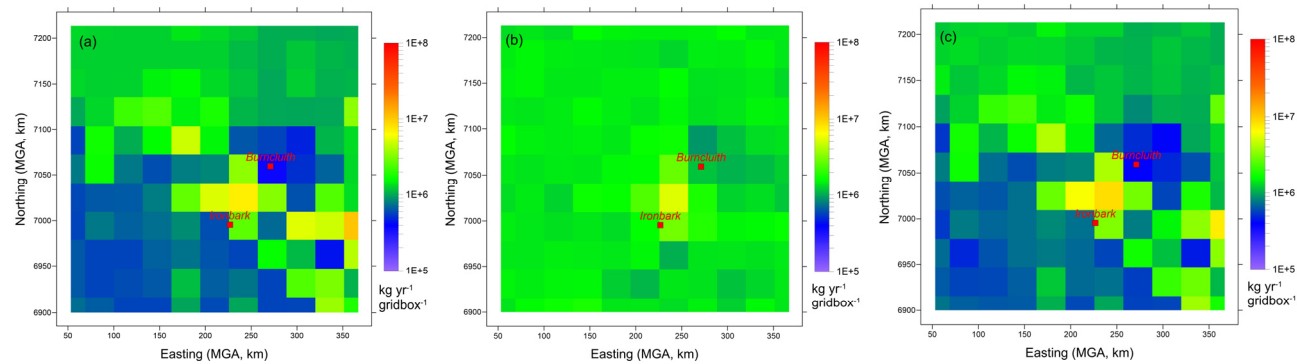

**Figure 9. Emission rates of CH₄ (kg yr⁻¹ gridcell⁻¹) (a) based on the bottom-up inventory, (b) estimated by the synthetic inversion using a uniform Gaussian prior with an uncertainty of $\sigma_p$ = 5% of the mean for each source, and (c) estimated by the synthetic inversion using the bottom-up inventory as a Gaussian prior with an uncertainty of $\sigma_p$ = 5% of the mean for each source. There are 11 × 11 sources, and the grid cell size is 31 × 31 km².**

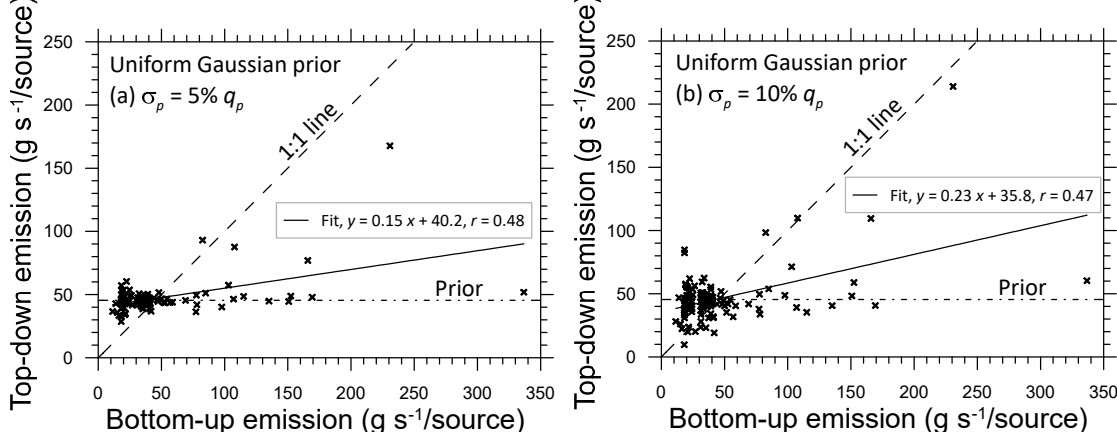

**Figure 10. Scatter plot of the bottom-up inventory methane emission rates (g s⁻¹ per source) versus those inferred from the inverse (top-down) methodology for the synthetic case involving a uniform Gaussian prior with a prior uncertainty of (a) $\sigma_p$ = 5% and (b) $\sigma_p$ = 10% of the mean for each source. The number of sources is 11 × 11. The dash-dot line is the mean value of the prior, the dashed line is the 1:1 line (i.e. perfect agreement) and the solid line is the least-squares fit.**

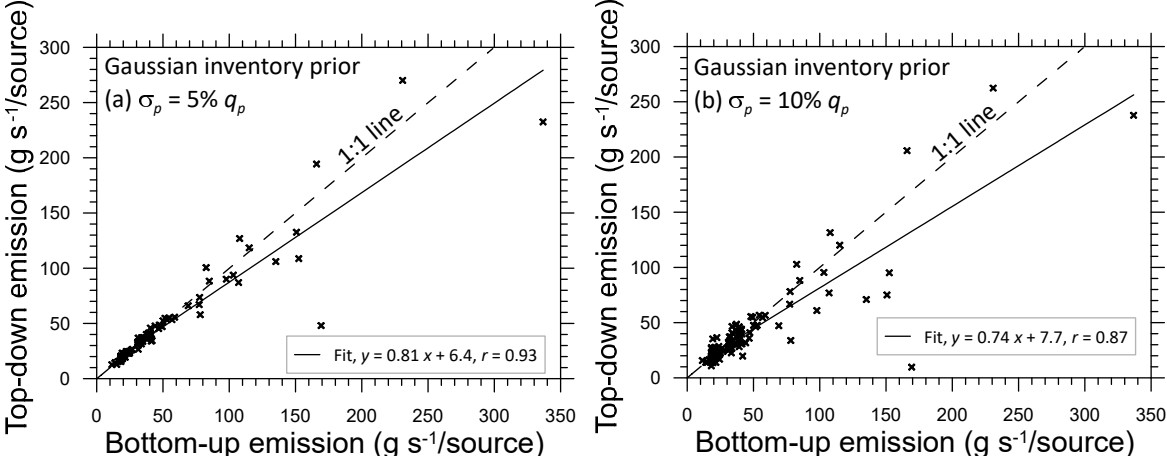

**Figure 11. Scatter plot of the bottom-up inventory methane emission rates (g s⁻¹ per source) versus those inferred from the inverse (top-down) methodology for the synthetic case involving the bottom-up inventory source emissions as the mean of a Gaussian prior with a prior uncertainty of (a) $\sigma_p$ = 5% and (b) $\sigma_p$ = 10% of the mean for each source. The number of sources is 11 × 11. The dashed line is the 1:1 line (i.e. perfect agreement) and the solid line is the least-squares fit.**

## 7 Inversion using the methane measurements

We now use the filtered methane measurements from the two monitoring stations to quantify emissions (with $N_m = 10581$, $\sigma$ = 3.5 ppb and $\sigma_m$ = 20%). The above synthetic case results have revealed that a good, tight prior is needed to infer emissions within the selected domain using concentrations from the two monitoring locations. We consider several cases to examine how the source inference is influenced using the real-world measurements depending on the type of prior that may be available, ranging from a non-informative one to the most informative we have, i.e. the bottom-up inventory.

### 7.1 Priors and inferred emissions

Three cases involving different priors are considered.

### 7.1.1 Non-informative uniform prior (Case 1)

A case of non-informative prior is first considered in which the only constraint is that the emission rate for each source lies within the broad range 10–10,000 g s⁻¹ with uniform probability, where the upper limit is nearly double the total domain-wide bottom-up inventory.

The inferred emissions (Figure 12a) between the two monitoring sites and around the centre of the region are qualitatively in accordance with the bottom-up inventory emissions (Figure 9a), but with larger magnitudes. In contrast, the inverse estimates in locations farther from these source areas are smaller than the inventory emissions. Notably, the total inferred emission with

the non-informative prior is $162.0 \times 10^6$ kg yr$^{-1}$ which compares well with the inventory total. The largest emission rate of about 1100 g s$^{-1}$ per grid cell in Figure 12a is about 10% of the upper bound of the specified prior range.

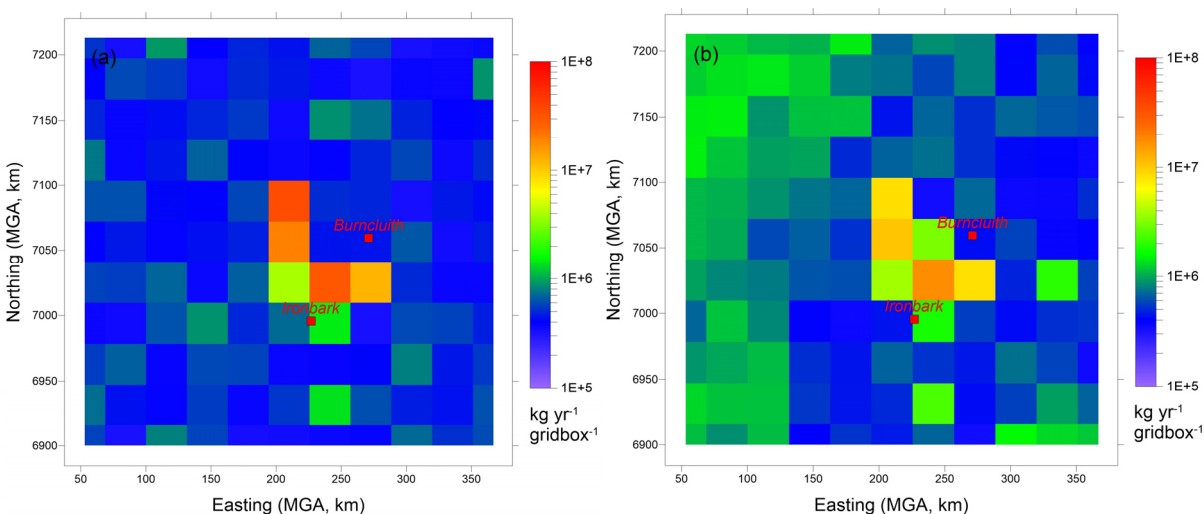

**Figure 12. Emission rates of CH$_4$ (kg yr$^{-1}$ gridcell$^{-1}$) estimated by the inversion: (a) with a non-informative uniform prior (Case 1); and (b) with a uniform Gaussian prior (Case 2). There are 11 × 11 sources, and the grid cell size is 31 × 31 km$^2$.**

### 7.1.2 Uniform Gaussian prior (Case 2)

Next, a more realistic prior PDF is specified with a Gaussian distribution having an identical mean of 45.4 g s$^{-1}$ and $\sigma_p$ = 10% of the mean, for each source. The mean is the same as that is used in one of the synthetic runs.

The inferred emissions for this case shown in Figure 12b are qualitatively similar to Figure 12a; however, in the former the high emission sources are relatively less pronounced, with emissions from other source locations generally being larger. The total annual emission from the Surat Basin obtained using this inversion is $143.1 \times 10^6$ kg yr$^{-1}$.

### 7.1.3 Gaussian prior using the bottom-up inventory emissions (Case 3)

In this case, as in the synthetic case corresponding to Figure 9c, the bottom-up inventory emissions (Figure 9a) are used in a Gaussian prior. A guided by the synthetic case results presented earlier, the uncertainty in the prior needs to be relatively small.

The inferred emission rates in Figure 13a obtained for Case 3 with $\sigma_p$ = 1% of the mean (Case 3a) appear very similar to the inventory emission rates (Figure 9a). The fact that even the intense emission on the eastern boundary of the domain present in the inventory is mostly reproduced despite this area being not sampled preferentially by the two network locations means that

the chosen prior with a very small uncertainty is somewhat too inflexible that forces the inversion towards a result that is very similar to the prior itself, thus likely overriding the information inherent in the concentration observations.

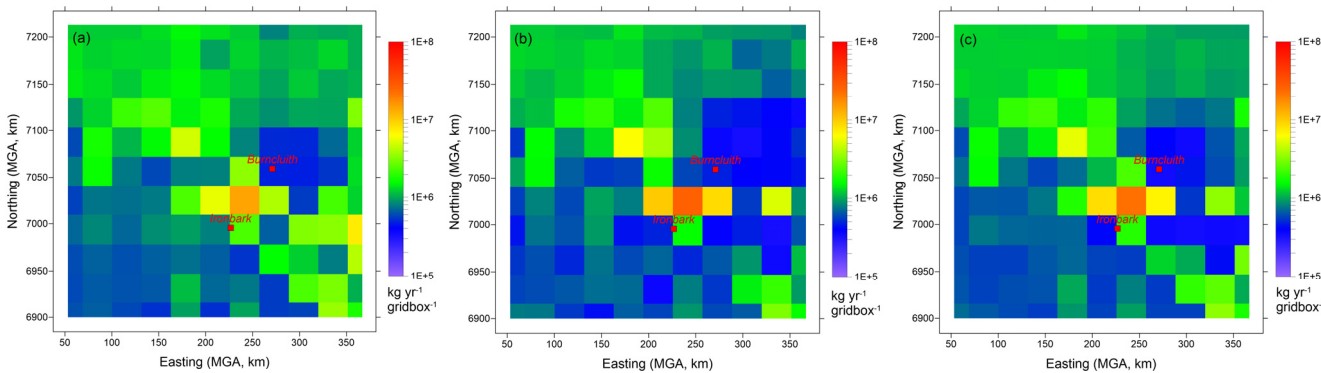

**Figure 13. Emission rates of CH₄ (kg yr⁻¹ gridcell⁻¹) estimated by the inversion with a Gaussian prior involving mean values equal to the bottom-up emissions (Figure 9a) and the standard deviation equal to (a) 1% (Case 3a), (b) 5% (Case 3b) and (c) 3% (Case 3c) of the mean values. There are 11 × 11 sources, and the grid cell size is 31 × 31 km².**

Figure 13b is the same as Figure 13a, except that the prior is relaxed by increasing $\sigma_p$ to 5% of the mean (Case 3b). This leads to the source areas in the centre of the Surat Basin and those between Ironbark and Burncluith becoming more conspicuous. In contrast, the source areas near the eastern boundary of the domain nearly fade, with the concentration observations applying greater influence in areas where the source-receptor relationship, shown in Figure 8, is stronger. Clearly, the inversion is sensitive to $\sigma_p$, however, it is apparent that a $\sigma_p$ between 1% and 5% of the mean would yield a reasonable trade-off between the benefit of the inversion approaching the prior in areas where the chances of the two monitoring stations detecting methane signal is small and simultaneously making sure that the selected prior would not unduly overrule the information supplied by the concentration measurements. Consequently, another inversion was performed for $\sigma_p$ = 3% of the mean (Case 3c). The inferred emissions from this run presented in Figure 13c in essence stand between those for $\sigma_p$ = 1% and 5% of the mean. This Case 3c inversion is our best estimate, which gives an annual total CH₄ emission of 165.8 × 10⁶ kg yr⁻¹. The fine tuning of the prior uncertainty is also guided by the need that the inferred emissions are able to describe the measured concentrations when used in a forward model simulation (see the validation Section 7.2).

Figure 14a presents the difference between the inferred methane emissions given in Figure 13c and the bottom-up inventory emissions in Figure 9a. The largest difference is found for the grid box between Ironbark and Burncluith, with the inferred emissions (22.9 × 10⁶ kg yr⁻¹) being larger by approximately a factor of three than the latter (7.3 × 10⁶ kg yr⁻¹). The total inventory emission for this source grid is controlled by CSG Processing (51%); feedlots, poultry and piggeries combined (32%); and CSG Production (6%) sectors.

The calculated posterior uncertainty (standard deviation) relative to the inferred mean emissions (%) corresponding to Figure 13c (Case 3c, $\sigma_p$ = 3% of the prior mean) is presented in Figure 14b. Most of these values are similar to the relative uncertainty in the prior (i.e. $\sigma_p$ = 3% of the prior mean). Interestingly, the farthest grid point due east of Ironbark, which corresponds to a relatively strong coal mine source in the bottom-up inventory (Figure 3d), has a disproportionally large uncertainty (~ 25%), probably due to limited sampling.

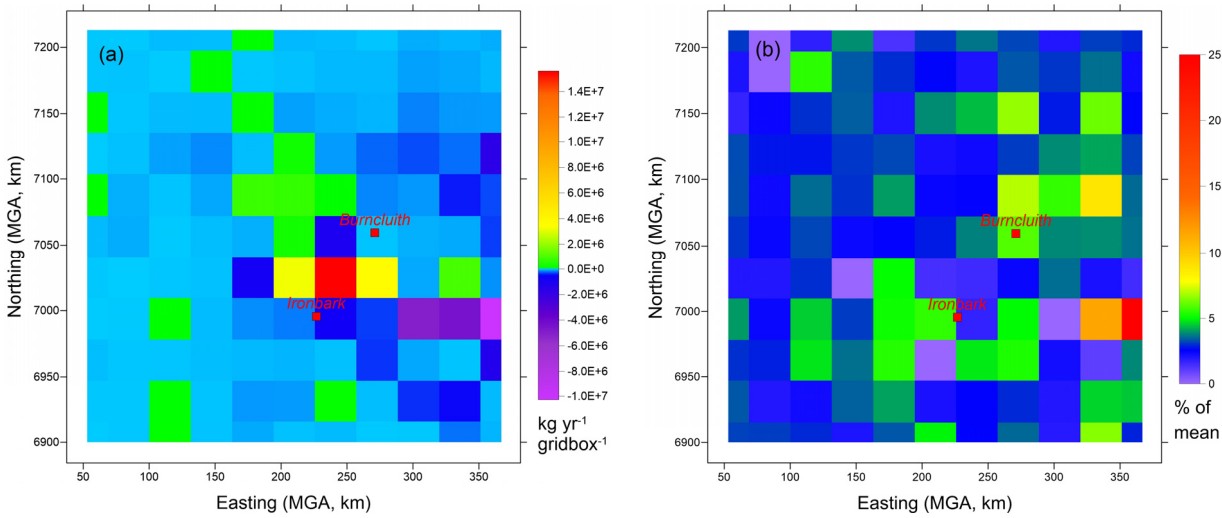

**Figure 14. (a) Difference between the Case 3c inferred methane emissions (Figure 13c) and the bottom-up inventory emissions in Figure 9a (kg yr⁻¹ gridbox⁻¹), and (b) posterior uncertainty (standard deviation) relative to the Case 3c inferred mean emissions (%). There are 11 × 11 sources, and the grid cell size is 31 × 31 km².**

## 7.2 Validation of the inferred emissions

To examine to what extent the inferred emissions represent the methane concentration measurements compared to the bottom-up emissions, we conducted three separate forward transport model runs using the inferred emissions from the above inversion Cases 1, 2, and 3.

The q-q plots for the Case 1 inferred emissions (Figure 15a, d) show that there is an overestimation of methane at both monitoring stations for the higher-end concentrations, but the simulated $CH_4$ at Ironbark is much better reproduced than when using the bottom-up emissions (grey lines). For Burncluith, the overestimation is almost as large in magnitude as the underestimation obtained when the inventory emissions are used.

The Case 2 inferred emissions obtained with a better prior lead to a significant improvement in the methane simulation, especially at Burncluith (Figure 15b, e).

As apparent from Figure 15c, f, the use of the bottom-up inventory as the prior in Case 3c with 3% prior uncertainty relative to the mean yields emission estimates that further improve the simulation of methane, especially at Ironbark. Comparatively, the use of 1% prior uncertainty leads to a better performance at Ironbark but worse at Burncluith. With 5% prior uncertainty, the performance is other way round. With the exception of about 4 outlying data points at the higher-end of the concentration distribution, the Case 3c inversion with 3% prior uncertainty (corresponding to Figure 13c) leads to the best overall model

reproduction of the measured CH$_4$ from the two monitoring sites. The underprediction seen when the inventory emissions are used (grey curves in Figure 15) is nearly eliminated.

Table 1 presents performance statistics for the three Case 3 inversions and for the bottom-up emissions as to how well they describe the methane concentration measurements at the two sites when used in the forward modelling. The observed ($O$) and modelled ($M$) concentrations are paired in time for these statistics, which are: $r$ = correlation coefficient, IOA = index of

agreement, $a$ = slope and $b$ = intercept of the linear best fit line (with observations along the x-axis), FB = fractional bias, and RMSE = root mean square error. FB $= 2(\bar{O} - \bar{M})/(\bar{O} + \bar{M})$, which varies between -2 (overestimation) and +2 (underestimation); and IOA $= 1 - [\overline{(M - O)^2}/\overline{(|M - \bar{O}| + |O - \bar{O}|)^2}]$, where 0 = no agreement and 1 = perfect agreement. The IOA, unlike $r$, is sensitive to differences between the observed and model means as well as to certain changes in proportionality.

Compared to the case with the bottom-up emissions, the inferred emissions improve the prediction of methane concentration at Ironbark, except for a slight decrease in correlation. At Burncluith, the improvement is limited to the slope. Note that these statistics are dominated by lower-end concentrations which are much more numerous than the higher-end concentrations. The q-q plots in Figure 15 on the other hand tend to emphasise more model performance for a relatively small number of higher-end concentrations.

Some deterioration in the model performance when the inferred emissions are used could be caused by the 11 × 11 source distribution representing the emissions in the domain being rather coarse (compared to 69 × 69 used for the bottom-up emissions). Considering the performance statistics in Table 1 and the q-q plots in Figure 15c and f, the Case 3c inversion is our best estimate of emissions.

**Table 1: Performance statistics for the emissions from the Case 3 inversions and for the bottom-up emissions as to how well they describe the methane concentration measurements at the two sites when used in forward modelling (*r* = correlation coefficient, IOA = index of agreement, *a* = slope, *b* = intercept, FB = fractional bias, RMSE = root mean square error).**

| Emissions | Ironbark ($N = 6432$) | | | | | | Burncluith ($N = 4149$) | | | | | |
|---|---|---|---|---|---|---|---|---|---|---|---|---|
| | *r* | IOA | *a* | *b* (ppb) | FB | RMSE (ppb) | *r* | IOA | *a* | *b* (ppb) | FB | RMSE (ppb) |
| Case 3a ($\sigma_p = 1\% \, q_p$) | 0.53 | 0.68 | 0.36 | 1153 | $0.61 \times 10^{-3}$ | 25.5 | 0.69 | 0.82 | 0.71 | 527 | $-0.45 \times 10^{-3}$ | 11.1 |
| Case 3b ($\sigma_p = 5\% \, q_p$) | 0.49 | 0.66 | 0.55 | 863 | $-1.98 \times 10^{-3}$ | 32.0 | 0.58 | 0.71 | 0.87 | 244 | $-1.26 \times 10^{-3}$ | 16.8 |
| Case 3c ($\sigma_p = 3\% \, q_p$) | 0.51 | 0.68 | 0.48 | 954 | $-0.72 \times 10^{-3}$ | 28.4 | 0.63 | 0.76 | 0.79 | 381 | $-0.86 \times 10^{-3}$ | 14.0 |
| Bottom-up inventory emissions | 0.57 | 0.59 | 0.25 | 1360 | $3.36 \times 10^{-3}$ | 25.4 | 0.74 | 0.84 | 0.61 | 707 | $0.35 \times 10^{-3}$ | 9.4 |

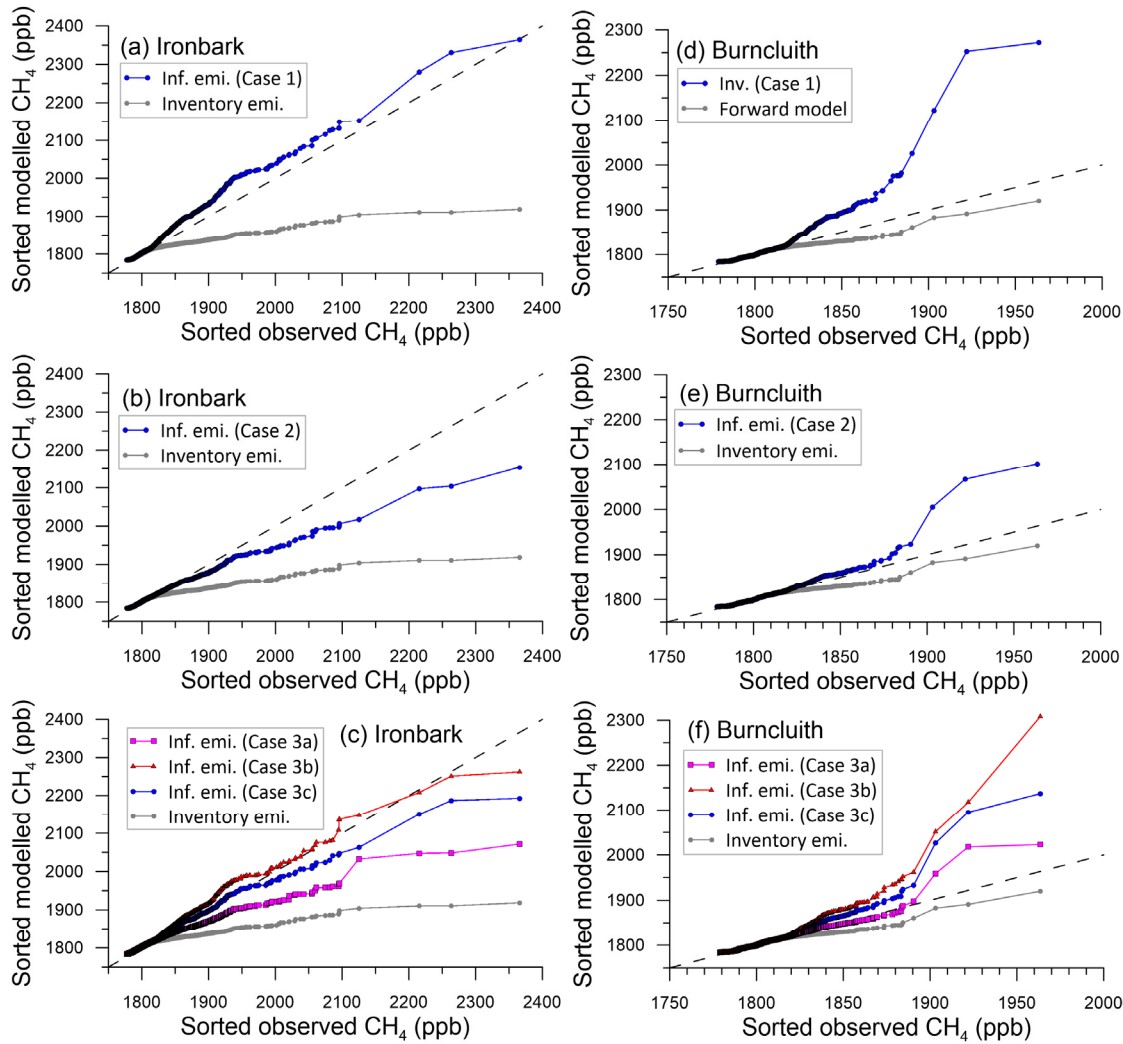

**Figure 15. Q-q plots showing the sorted hourly observed versus the sorted modelled CH₄ at the Ironbark and Burncluith monitoring stations. The forward modelled concentrations utilise emission estimates from the Case 1, Case 2, and Case 3 inversions. The forward model concentrations obtained using the bottom-up emissions are also shown (grey line). The dashed 1:1 line represents perfect agreement.**

## 7.3 Emissions from the CSG area

Given the focus on CSG activity related emissions in the Surat Basin, we compare the aggregate bottom-up and inferred emissions from the CSG areas, many of which are concentrated near and between the two monitoring stations. The subdomain that includes all the CSG sources in the study area is shown Figure 16, which is an area of about 18260 km², 15% of the study domain, and covers 19 of the 121 source grids considered. The CSG subdomain also contains emissions from other sectors (see Figure 3).

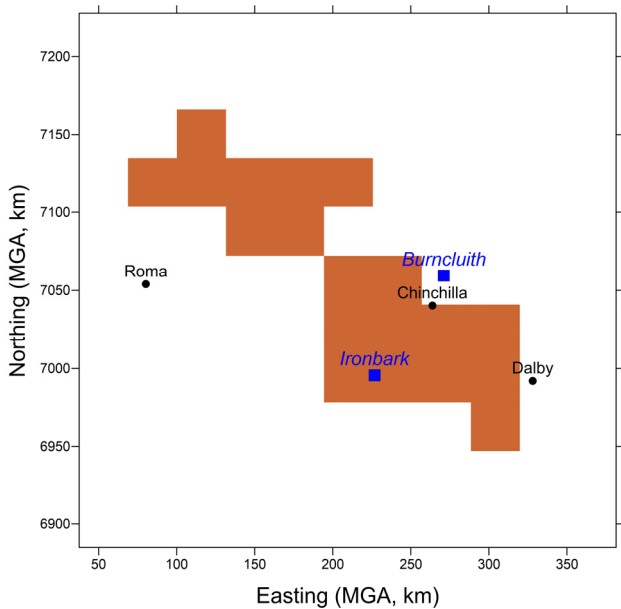

**Figure 16. Subdomain of the study area that covers all the CSG source areas (shaded grid cells) included in the bottom-up emission inventory. It covers 19 of the 121 source grids (each with a source footprint of 31 × 31 km$^2$) considered in the inverse modelling.**

The total bottom-up inventory emissions from the CSG sub-domain is $47.7 \times 10^6$ kg yr$^{-1}$ (cf. $173.2 \times 10^6$ kg yr$^{-1}$ for the study domain) whereas that obtained using the inversion (Case 3c, Figure 13c) is $63.6 \times 10^6$ kg yr$^{-1}$ (cf. $165.8 \times 10^6$ kg yr$^{-1}$ for the study domain) which is 33% larger than the former. The total bottom-up emission for this subdomain is dominated by CSG (34.7%, of which 30.6% is due to CSG Processing), followed by grazing cattle (29.9%), feedlots (23.5%) and coal mines (7.7%), which together account for 95.8% of the emissions from this area. Since the inverse methodology does not differentiate

between source sectors, emissions from individual sectors cannot be inferred. Considering that the grazing cattle emissions are diffuse sources and thus not likely responsible for peaks in the measurements that dominate the inverse estimates, and since feedlots are scattered throughout the domain (Figure 3c) including the non-CSG areas from where there is no general inference of higher emissions, it is plausible that the increase in the inferred emissions would mainly correspond to CSG as the source sector.

A considerable portion of the CSG emissions is in the area between the two monitoring stations. The inferred emissions in this area are much greater than the corresponding bottom-up inventory emissions. However, this area also has significant coal mining emissions nearby (Figure 3d) and it is possible that the methane emissions from a combination of these two source sectors are much larger than the inventory emissions.

Conversely, the total bottom-up inventory emissions from the rest of the study domain (i.e. the non-CSG subdomain) is 125.5

$\times 10^6$ kg yr$^{-1}$, whereas that obtained using the inversion from Case 3c is $102.2 \times 10^6$ kg yr$^{-1}$ which is 18.5% lower than the

former. The total bottom-up emission for the non-CSG area is dominated by grazing cattle (62.7%), followed by feedlots (24.8%) and coal mines (8.6%), which together account for 96.1% of the emissions from this area. It is possible that the emission factor of 84 kg $CH_4$ animal$^{-1}$ yr$^{-1}$ for Australian grazing cattle (Harper et al., 1991) used in the bottom-up inventory (see the Supplement S6) is an overestimate (cf. 51 kg $CH_4$ animal$^{-1}$ yr$^{-1}$ for beef cattle (pasture) used by the Australian National Inventory Report (NIR, 2017) or 63 kg $CH_4$ animal$^{-1}$ yr$^{-1}$ for non-dairy cattle for the Oceania (IPCC, 2019)), and that would be consistent with a lower top-down methane emission from the non-CSG area compared to the inventory. This also means that the CSG component of the top-down emissions in CSG sub-domain could be higher to compensate for the lower grazing cattle emissions if a lower emission factor for grazing cattle is used.

Apart from the uncertainties associated with the bottom-up emissions, potential methane emissions from some sources, namely wetlands (the amount of which in the area is very limited; https://wetlandinfo.des.qld.gov.au), land clearing, termites, material handling and fuel usage related to mining activities, ground-water wells, and biomass burning are not part of the bottom-up emissions. In contrast, all $CH_4$ sources are implicitly represented in the inversions, apart from the biomass burning events which have been filtered out using the CO filter. It is difficult to pinpoint which source sectors might be underrepresented in the bottom-up inventory without some kind of source discrimination, for instance, through the use of tracers such as the $CH_4$ isotopes.

### 7.4 Temporal variation of the inferred emissions

Here we apply the inverse model with the Case 3c settings (as used for Figure 13c with 3% prior uncertainty relative to the mean) to 3-monthly measurement blocks within the measurement period (July 2015–December 2016) in order to examine potential temporal variation of the inferred emissions, bearing in mind that for a 3-monthly simulation the amount of concentration data supplied to the Bayesian inversion is much less than that for the full simulation. Figure 17a presents the 3-monthly variation of the inferred emissions as kg $CH_4$ yr$^{-1}$ (bar plots), along with the time invariant bottom-up inventory emissions (red line) and inferred emissions from Case 3c (blue line). The 3-monthly emission rates are within 165–180 kg yr$^{-1}$ and are generally larger than the time invariant inferred emissions obtained using the measurements from the full period. We believe that this is at least partly because as the amount of information supplied to the inversion reduces, the inferred emissions are not modulated to the same extent as that for the full period, and thus they tend to move closer to the bottom-up inventory which is used as a prior. Another related reason could be the narrowing of the amount of source area represented by the source-receptor relationship because of seasonal winds falling in relatively narrow directional sectors compared to the broader wind rose for the full period.

Figure 17b is the same as Figure 17a but for the CSG subdomain. The 3-monthly inferred emissions lie between the bottom-up inventory value and the time invariant inferred value. Again, as in Figure 17a, 3-monthly inferred emissions push towards the inventory value.

Figure 17c is the same as Figure 17a but for the non-CSG subdomain (which is dominated by grazing cattle emissions (62.7%) as per the bottom-up inventory). In this plot, we also present a 3-monthly climatological average (1992 – current 2020) of rainfall at the Dalby airport (location 27.16°S, 151.26°E), located next to the town of Dalby, within the study domain. The

775 rainfall data were obtained from the Australian Bureau of Meteorology (from http://www.bom.gov.au/climate/averages/tables/cw_041522.shtml). There is a good correlation ($r = 0.79$) between the 3-monthly inferred non-CSG methane emission and the rainfall, suggesting that the inferred emission variation could, to some extent, be attributed to the seasonality of rainfall which would influence areas such as pasture growth and thus methane emissions from grazing. This correlation for the 3-monthly inferred emissions for the full domain (Figure 17a) is 0.71 and it is

780 -0.06 for those from the CSG subdomain (Figure 17b). It is reasonable to assume that the higher the rainfall the higher the grazing cattle emissions, and in that case these $r$ values indicate that the seasonal variability in the inferred emissions within the full domain is, to a lesser degree, also influenced by such emissions. However, the inferred emission seasonality within the CSG area does not correlate with rainfall, meaning that the emission seasonality is possibly dominated by the CSG sources.

Another potential contributor to the temporal variability in the inferred emissions in Figure 17 is the seasonality of the winds
which influences the source-receptor relationships.

The uncertainties in the inferred seasonal emissions Figure 17 is around 5% of the mean – a relatively small value largely the result of a tight prior.

To test how well the temporal variation of the inferred emissions represents reality, we conducted a forward TAPM run using these emissions , and the resulting q-q plots (red dots) are shown in Figure 18. The methane data at Burncluith are best described

by these 3-monthly varying emissions compared to any other emission setup, but at Ironbark, these emissions underestimate the data (the inversion setup corresponding to Figure 15c best describes the Ironbark data).

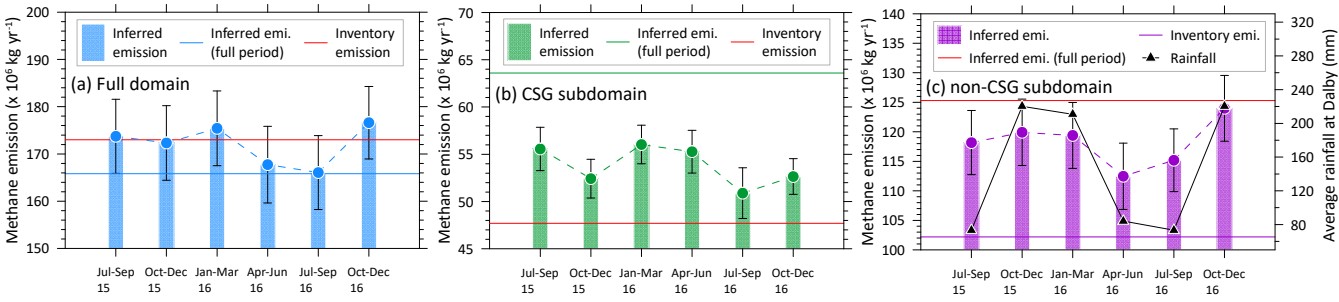

**Figure 17. 3-monthly variation of the inferred emissions (bar plots), including one standard deviation uncertainty (~ 5% of the**
795 **mean), for (a) the full study domain, (b) the CSG subdomain, and (c) the non- CSG subdomain. The respective time-invariant bottom-up inventory emissions (red line) and the time invariant inferred emissions from the Case 3c inversion (Figure 13c) are also shown. Note the emission units. In (c), a 3-montthly climatological average (1992 – current 2020) of rainfall at the Dalby airport located within the study domain is also shown.**

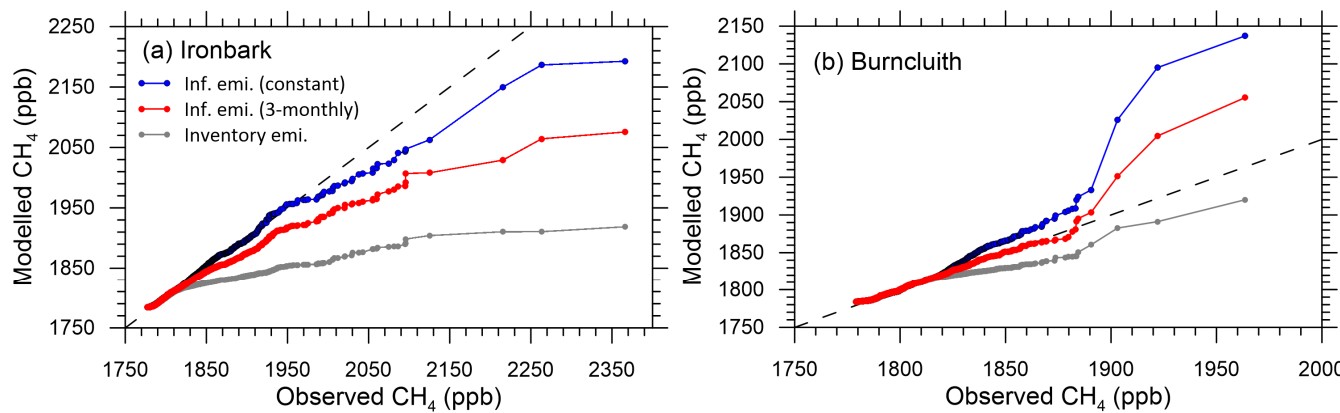

**Figure 18. Q-q plots showing the sorted hourly observed versus the sorted modelled CH₄ at the two monitoring stations. The modelled concentrations are predicted using: the time-invariant inferred emissions from the Case 3c inversion (with 3% uncertainty in the prior relative to the mean) (blue dots); the 3-monthly inferred emissions (red dots); and the bottom-up inventory emissions (grey dots). The dashed 1:1 line represents perfect agreement.**

Given the rapid rise in the CSG production in the Surat Basin, one may deduce that the 2016 CSG methane emissions were larger than the 2015 bottom-up emissions and, therefore, could potentially explain the top-down emissions in the CSG area being higher than the inventory emissions. Figure 19 shows that compared to July–December 2015, the total CSG produced was higher by 32% during January–June 2016 and by 45% during July–December 2016, which correlates with an increase in the number of CSG production wells in the area. However, Figure 19 also shows that there is a downward trend in the amount of flared/vented gas. Considering, based on the bottom-up inventory in Section 3, that venting (from processing) is the biggest contributor (88%) followed by flaring (8%) (from both processing and production) to the total CSG methane emissions, it is plausible that despite the increase in the CSG development in the area the CSG-related methane emissions have not increased, and that they may have even gone down. The temporal variation of the inferred emissions in Figure 17b for the CSG dominated area also does not indicate any consistent increase in emissions from 2015 to 2016. Thus, the 33% higher top-down emission estimate from the CSG area compared to the inventory estimate cannot be explained in terms of the growth in the CSG production from 2015 to 2016 and is possibly related to underestimated or missing emissions in the inventory. This also implies that the emissions from CSG may be more closely related to practices in the industry than to the amount of CSG produced.

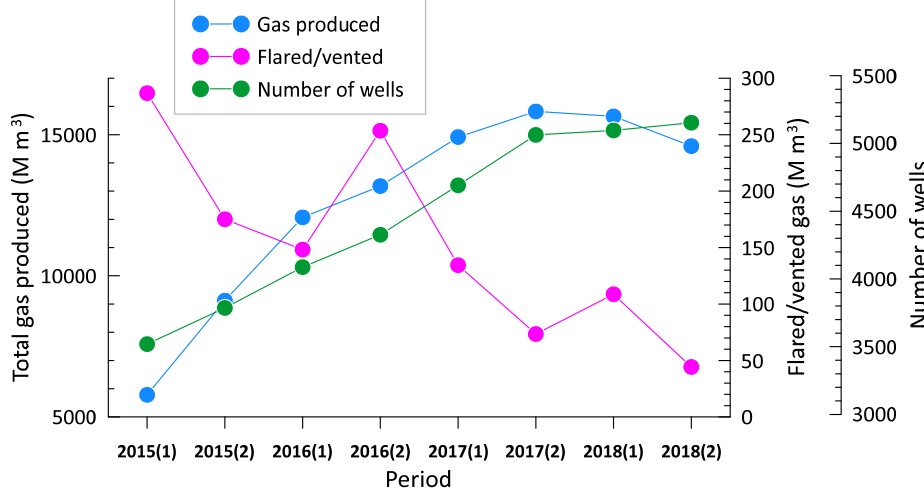

**Figure 19. Six-monthly trends of the total CSG produced, amount of flared/vented gas, and number of wells in the Surat Basin (data from https://www.data.qld.gov.au/dataset/petroleum-gas-production-and-reserve-statistics[1]).**

## 7.5 Sensitivity of inversion to background methane

Figure 4 shows that there is a slight difference in the estimated background $CH_4$ levels between the two monitoring locations, with the Ironbark background methane larger by 1 ppb on average than Burncluith and the standard deviation of the background differences being 1.4 ppb, the latter is comparable to the background concentration uncertainty (= 3.5 ppb) considered in the inversion.

We conducted an inversion sensitivity test with the same model setup as that for Figure 13c (Case 3c), except that instead of using the background times series that was averaged over the two sites we used the respective background time series for the two sites. The results were virtually the same compared to Figure 13c, other than some insignificant changes in areas with low emissions. Table 2 gives the annual inferred emissions, which show no sensitivity.

Our background methane calculation methodology (Supplement S3) assumes that under very vigorous atmospheric mixing
conditions in the daytime, the measured concentrations within study domain represent methane levels both within and outside the domain boundaries, so that the measured concentrations can be taken to represent the background under such conditions. Because the background concentration is calculated from the measurements within the source region under study, there is a possibility that the real background is potentially lower than what we have used. To examine this, another inversion sensitivity

---

[1] This data file places the gas fields of Spring Gully and Peat within the Bowen Basin whereas in our bottom-inventory these are part of the Surat Basin. This is because of how the gas field zones and basin boundaries are defined. The gas fields included in our study are based on their geographic locations relative to the square study domain selected. Adding these two gas fields to the Surat Basin does not change the trends shown in Figure 19.

test was conducted by using an alternate methane background times series (with all other settings the same as the final Case 3c inversion) and this is described in detail in the Supplement S5. Essentially, the alternate background was constructed using the original averaged background from the two sites and the marine baseline methane measurements from the Cape Grim Baseline Air Pollution Station (https://capegrim.csiro.au), located on the north-west tip of Tasmania (40.7ºS, 144.7ºE). The marine baseline methane represents concentration levels without the direct influence of the continental sources. The alternate background falls between the average Surat background as used in the paper and the Cape Grim baseline and is, on average, lower than the original Surat background by 2.8 ppb. (On average, the Cape Grim marine baseline was 8.4 ppb lower than the original Surat background used).

The inversion results in Table 2 show that compared to the inferred emissions obtained using the original background methane the alternate background gives total emissions that are 6.8% higher, while the increase is smaller at 3.9% in the CSG subdomain and larger at 8.5% in the non-CSG region. The overall increase is expected because the increase in the measured concentrations by 2.8 ppb as a result of the use of the alternate background needs to be accounted for by the inversion by enhancing the amount of inferred emissions. We also find that the amount of increase in the inferred emissions with the alternate background is almost uniformly spread through the study domain relative to the total emission, and that there are no significant spatial distributional shifts in the inferred emissions with the two background choices.

There are possibly other and better ways of calculating the background methane concentration, such as having methane measurements at many locations around the perimeter of the study domain (which is often subject to operational and budget constraints) or modelling methane at much larger scale, preferably global, with data assimilation, which could then provide concentration boundary conditions needed for the regional modelling.

**Table 2: Inferred emissions ($\times 10^6$ kg yr$^{-1}$) obtained using: the methane background averaged over the two sites (as used in the paper, Case 3c), the individual methane background from the two sites, and the alternate methane background calculated using the Cape Grim baseline methane data (see Supplement S5). The values in the parentheses are % change over the inferred emissions using the averaged background. The bottom-up inventory emissions are also included for comparison.**

| Selected background | Total | CSG subdomain | Non-CSG subdomain |
|---|---|---|---|
| Average background (as used in this paper) | 165.8 | 63.6 | 102.2 |
| Separate backgrounds from the two sites | 164.8 (-0.6%) | 62.7 (-1.4%) | 102.1 (-0.1%) |
| Alternate background (see Supplement S5) | 177.0 (+6.8%) | 66.1 (+3.9%) | 110.9 (+8.5%) |
| Bottom-up inventory emissions | 173.2 (+4.5%) | 47.7 (-25%) | 125.5 (+22.8) |

**8 Conclusions**

This paper presented quantification of methane emissions from the CSG producing Surat Basin, an area of $350 \times 350$ km$^2$ in Queensland, Australia. The 2015 bottom-up methane emission inventory served as a very useful prior in our regional inverse methodology based on a Bayesian inference approach that utilised hourly-mean CH$_4$ concentrations monitored at the Ironbark and Burncluith stations for 1.5 years, hourly source-receptor relationship, and an MCMC technique for posterior PDF sampling.

The largest contribution to the emissions in the bottom-up methane inventory was from grazing cattle (~50%), cattle feedlots (~25%), and CSG processing (~8%), with the aggregate emissions in the study area being approximately $173.2 \times 10^6$ kg CH$_4$ yr$^{-1}$. Although the forward transport modelling with the bottom-up emissions yielded a credible simulation of the suitably filtered observed methane concentrations, about 15% of the higher-end concentration observations were underestimated.

The importance of specifying a suitable prior in the Bayesian inference was made apparent by the synthetic inversion, 875 demonstrating the use of the bottom-up inventory with a narrow uncertainty as being a good choice for that purpose when only two monitoring locations available. For inversion with the real methane measurements, a Gaussian prior having mean values taken the same as the bottom-up emissions with an uncertainty equal to 3% of the mean yielded the best emission distribution, as evident from its performance in faithfully reproducing the measured methane concentration time series. This inverse setup yielded a domain-wide emission of $165.8 \times 10^6$ kg CH$_4$ yr$^{-1}$ which is very slightly less than the one obtained 880 from the bottom-up inventory. However, within a subdomain covering all the CSG source areas, the inferred emission $63.6 \times 10^6$ kg CH$_4$ yr$^{-1}$ is 33% larger than that deduced from the bottom-up inventory. The dominant localised inventory emissions in this area are from CSG, followed by feedlots. Since feedlots are scattered throughout the domain including the non-CSG areas from where there is no indication of higher emissions, it is plausible that the increase in the inferred emissions would mainly correspond to CSG as the source sector.

Despite the amount of concentration data going into the seasonal inversion being relatively limited, the inferred seasonal variation of methane emissions from the non-CSG subdomain correlated well with climatological seasonal rainfall in the area, suggesting a possible link with the seasonality of agricultural emissions. This correlation was almost zero for the CSG subdomain, possibly due to the CSG sources dominating the seasonality.

There was some sensitivity to the background methane concentration observed in the inversion, and we believe that further 890 approaches to the background calculation are necessary for regions like the Surat Basin.

The source-receptor relationship showed that having only two monitoring stations is inadequate for sampling distant source areas within the large study domain, especially areas in the south-east and north-west corners (the network design for the two monitoring stations mainly focused on the central CSG regions). Lengthening the measurement period to sample these areas better would not have helped because the wind climatology of the area is not likely to change considerably. When source areas 895 are not sampled well, one may impose stricter priors that are more credible than the inferred emissions, or alternatively increase

the number of stations. The former strategy is probably reflected in our use of a small uncertainty in the prior (i.e. 3% of the mean) for the best inversion case. A smaller prior uncertainty pushes the inversion more towards the prior itself with distant source areas not sampled sufficiently by the network sites looking like the prior distribution.

The inverse methodology could not distinguish between different source categories, mainly because the concentration of methane alone was monitored and not tracers specific to methane source types. To do source discrimination and attribution, monitoring of tracer species such as methane isotopes ($^{13}CH_4$, $CH_3D$ and $^{14}CH_4$), or other hydrocarbons in cases where they are associated with the source gas, would prove useful when suitable sampling systems or instrumentation for field deployment become available.

The methods developed in this study could be used to improve the monitoring and management of greenhouse gas and other air emissions from the onshore gas industry, including that in the Surat Basin. They provide independent information to industry and communities living in gas development regions on one of the main environmental impacts potentially arising from onshore gas developments. Improved quantification of methane emissions on the regional scale is an important step in emissions reductions from the onshore gas sector and possibly other industries. The present top-down method is particularly suited to distributed emissions with potentially unknown locations across a large geological gas reservoir and gas production infrastructure. If monitoring is deployed before gas exploration and production begins then a baseline would be established from which emissions from the industry might be detected. Ongoing top-down quantification, with monitoring stations located close to where emissions appear and with source-specific information from tracers could provide the information necessary to validate emissions from the gas industry to support greenhouse gas inventories.

### Data availability

The data and model output included in this paper can be made available by contacting the corresponding author (Ashok Luhar: ashok.luhar@csiro.au).

### Author contributions

AKL performed the model development and application, analysed model output and data, and wrote the paper with contributions and comments from the co-authors. DME conducted the field study design, in-situ monitoring and data analysis, and GISERA project management. ZML conducted the in-situ monitoring, data collection, and data analysis. JN contributed to data processing. DS assisted with the monitoring sites, instrumentation and data collection. LS developed the bottom-up inventory. CO provided general information on methane sources.

**Competing interests**

The authors declare that they have no conflict of interest.

**Acknowledgments**

This work was partially supported by CSIRO's Gas Industry Social and Environmental Research Alliance (GISERA) (https://gisera.csiro.au) and Active Integrated Matter (AIM) Future Science Platform (FSP) (https://research.csiro.au/aim). Mark Kitchen and Steve Zegelin provided valuable instrumental and technical support. Staff of GASLAB at CSIRO Oceans and Atmosphere, Aspendale, provided the atmospheric composition measurement support and calibration standards required

for this work. The landowners of the Burncluith and Ironbark sites provided essential support of the monitoring. The authors thank Peter Rayner, Martin Cope and Dimitri Lafleur for their helpful comments on this work, and Natalie Shaw for her assistance with the preparation of the bottom-up inventory for the Surat Basin. Damian Barrett advised on this work, and Stuart Day furnished insights into local source monitoring. Useful comments by the two anonymous referees and Bryce Kelly are much appreciated. NCEP Reanalysis data provided by the NOAA/OAR/ESRL PSL, Boulder, Colorado, USA, from their Web

site at https://psl.noaa.gov.

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
