# Peer review of "Quantifying methane emissions from Queensland's coal seam gas producing Surat Basin using inventory data and an efficient regional 5 Bayesian inversion"

_Atmospheric Chemistry and Physics, 2020_

## Short Comment (SC1) · 20 Jun 2020

Bottom-Up Inventory Base Values and Emission Factors

Comments on Luhar et al. (2020) "Quantifying methane emissions from Queensland's coal seam gas producing Surat Basin using inventory data and an efficient regional Bayesian inversion (https://doi.org/10.5194/acp-2020-337)

[A pdf version of these comments is attached with improved formatting]

[Figure]

This manuscript will make an important contribution to the ongoing scientific debate about bottom-up versus top-down greenhouse gas assessments. The authors should be congratulated for shifting the research focus in the Surat Basin, Australia, from locating methane sources to quantifying the rate of emissions from various sources. This is a valuable scientific contribution.

All my comments below relate to the bottom-up inventory which is used as a reference point for many of the discussions in Luhar et al. (2020) and as a prior for the regional Bayesian inverse model methane emission flux estimate. As documented in Luhar et al. (2020) there is a discrepancy between the top-down versus bottom-up modelling estimate for total methane emissions and apportioning to sub-areas within the domain of the study. I hope the comments below will assist in better methane source apportionment and that this will improve the alignment of the inventory with the inverse Bayesian modelling results.

As recently presented at EGU 2020 in Lu et al. (2020) UNSW researchers have developed their own bottom-up inventory in the Surat Basin for the year 2018. That presentation should convey to the authors of this manuscript that an updated bottom-up inventory for the Surat Basin will be shortly submitted for review (Kelly et al., in preparation). It would be constructive for the science of inventory collation if the inventory presented in Luhar et al. (2020) and the inventory in Kelly et al. (in preparation) converge on both workflow and methane emission estimates for the primary sources of methane. There will be a significant difference between the Luhar et al (2020) 2015 inventory and the 2018 inventory to be presented in Kelly et al. (in preparation), but those differences should be traceable (different amounts of gas produced, changes in the population of cattle etc).

From the insights in preparing the Kelly et al. (in preparation) inventory, and the airborne measurement observations in Neininger et al. (2020), I have a number of questions with respect to the inventory in Luhar et al (2020). I hope the comments will result in an improved match between the inventory and modelling in Luhar et al (2020), and

the development of a common inventory template for future studies.

Concerns with the lack of details provided on the inventory calculations

The essential bottom-up inventory details on base quantities and emission factors for Coal Seam Gas (CSG) production and processing are not presented in Luhar et al. (2020). Nor are there any details on CSG produced water volumes or management controls. As a stand-alone reference it is not currently possible to validate the data presented in Table 2 in Luhar et al. (2020). There are more details in Luhar et al. (2018), however when I tried to access Luhar et al. (2018) at https://publications.csiro.au/rpr/pub?pid=csiro:EP185211 the link to the report was broken (access attempts 14 to 20 June 2018, none successful). This highlights the importance of putting the core information used in Luhar et al. (2020) for the inventory in the supporting information. Why is Luhar et al. (2018) not cited in Luhar et al. (2020)?

Suggested manuscript revision inventory calculations

In the supporting information a table needs to be presented that lists the base quantity, emission factor used, clear referencing of the document(s) for the emission factor (and for each document clear referencing of the table and row selected for the emission factor), and justification for the selection of the emission factor, especially if it is not the default value as listed in either the 2006 IPCC Guidelines for National Greenhouse Gas Inventories (or 2019 refinement) or the Australia Government value as applied in Australia's National Greenhouse Gas Inventory (UNFCCC classifications). This is needed for all categories, Coal Seam Gas, Grazing Cattle, Feedlot Cattle, etc. Luhar et al. (2018) needs to be cited, as there is considerable overlap between that report and Luhar et al. (2020).

Points of clarity required with the CSG bottom-up inventory estimation of emissions

Because Luhar et al. (2020) does not adequately list the base CSG data I can only make a check on the inventory values presented using data in the public domain. All tallied gas volumes and produced water data for Queensland for each petroleum lease (Pel) in the Surat Basin are published online by the Queensland Government (https://www.data.qld.gov.au/dataset/petroleum-gas-production-and-reserve-statistics, accessed 18 June 2020). Luhar et al. (2018) does provide better information on most aspects of the inventory, but there are points in Luhar et al. (2020) that could be added to help all readers of the manuscript.

No listing of Pels covered is provided in Luhar et al. (2020). From the Queensland Government database in the Surat Basin gas was produced from 3519 wells in the period ending 30/06/2015 and 3768 wells in the period ending 31/12/2015. This total well number used in producing gas is actually slightly lower than reported in this manuscript (4628 wells * 0.85 = ~3934). Below I therefore use the complete production data for the Surat Basin (all from the Walloon Coal Measures).

It is well documented that there are emissions from the water management ponds in the Surat Basin: refer to Iverach et al. (2015) Figure 3, and Nisbet et al. (2020) Figure 10. There are no volumes reported for CSG produced water, nor any reference for the total emissions from produced water as an isolated category in either Luhar et al. (2018) or Luhar et al. (2020), and no reference is made to produced water control factor. There is no reference to produced water emissions due to CSG activities in Luhar et al. (2018) Table 15 Total Methane Emissions (kg/year). However, we can make a check of the likely emissions from produced water using the Queensland Government public domain data. In 2015 in the Surat Basin the volume of produced water was 48591.79 Mega litres. The API 2009 average water tank emission factor is 0.31955 tonnes CH4/1000 m^3 produced water (page 5-57, Table 5-10) (Also refer to NIR (2020) Volume 1 page 143 and Table 3.44, NGER Method 2 (API 2009)). Using the API 2009 emission factor, assuming a control factor of zero, up to 15,527,505 kg CH4/year of emissions is likely released from the "Produced Water". For the year 2015 the total gas produced in the Surat Basin was 14905.77 Mm^3. The amount flared and vented was 461 Mm^3, and 316.03 Mm^3 was used in production. Considering just total

production the API (2009) Table 6.2 Facility Level Average Fugitive Emissions Factors estimate for Production (Onshore gas production) emission factor is 9.184E-01 tonnes CH4/10ˆ6.mˆ3 produced, yielding a Surat Basin production estimate of 13,689,459 kg CH4/year. The API (2009) Table 6.2 Facility Level Average Fugitive Emissions Factors estimate for Gas Processing Plants is 1.032E+00 tonnes CH4/10ˆ6.mˆ3 processed, yielding a Surat Basin processing estimate of 15,382,755 kg CH4/year. This tallies to 44,599,719 kg CH4/year for Surat Basin CSG Processing, Production and Produced Water emissions. But from the original Luhar et al. (2018) report the CSG Processing and Production tally is 16,528,838 kg CH4.

I acknowledge that there can be many refinements to these estimates, but the estimates presented above are more in alignment with the Bayesian inverse modelling results presented in Luhar et al. (2020). Because the base quantities for CSG gas (produced, venting, flaring, and used in production) and CSG produced water are not listed in Luhar et al. (2018) and Luhar et al. (2020) we cannot begin to understand why the CSG production and processing bottom-up inventory methane emission estimates appear to be low in Luhar et al. (2018) and Luhar et al. (2020). For the bottom-up inventory reported in Luhar et al. (2020) to have any scientific merit the base quantities and emission factors used need to be presented.

Suggested manuscript revision CSG bottom-up inventory

In the revised manuscript it is recommended for the CSG inventory portion of this manuscript that the inventory table using the same categories reported by the Queensland Government in the Petroleum Gas Production and Reserves excel file be used, or a Table following the UNFCCC classifications be used. A table using either categories (classifications) would clearly separate emissions associated with Water Production. For example, a listing according to UNFCCC classifications would include: 1.B.2.b.2.i Water Production 1.B.2.b.2.ii Pipelines 1.B.2.b.2.ii Stations 1.B.2.c1 Venting 1.B.2.c2 Flaring etc

A complete listing of petroleum leases (Pels) used in this investigation needs to be added to the supporting information.

Points of clarity required with the cattle bottom-up inventory estimation of emissions

The choice of using Harper et al. (1999) for cattle emission factors needs to be justified. This is a respected reference with 109 citations in Scopus. However, it is neither the IPCC nor Australian Government recommendation. The grazing cattle emission factor used is this paper was established under artificial conditions, near Canberra (a very different climate to the Surat Basin), using rather old equipment compared to modern systems.

From Luhar et al. (2018) "Methane emission factor for grazing cattle of 0.23 kg CH4/animal/day based on direct measurements (Harper et al., 1999), which is 83.95 kg/CH4/head/year. The use of this emission factor contradicts the statement in Luhar et al. (2020) that "Standard methodologies were generally adopted with data from various State and Federal Government Departments (e.g. (National Pollutant Inventory (NPI), National Greenhouse and Energy Reporting (NGER), and National Resource Management (NRM)). "

The choice of implied emission factor for grazing cattle has a significant impact on the inventory. The Australian Government (NIR 2017, which reports for the year 2015) uses an implied emission factor of 51 for Beef Cattle – pasture (Table 5.11 Implied emission factors – enteric fermentation). Can the authors explain why they did not use the recommended value for Australia, or the IPCC default value of 60, or the Oceania default emission factor of 63 (IPCC 2019 Volume 4, Table 10.11)?

Using the Australian Government recommendation of 51 the total estimate is only 55,389,009 kg/year (51 * 1,086,059). But the Luhar et al. (2018) emission estimate is 92,991,979 for grazing cattle (this would require an emission factor of 85.62), which appears to be an overestimate for grazing cattle of 37,602,970 kg/year. Given that the category grazing cattle is the largest source of methane reported in Figure 2 (Luhar

et al 2020), some clarity on why Harper et al. (1999) was used to assign an emission factor for grazing cattle would address concerns that the grazing cattle emissions have been overestimated.

Suggested manuscript revision cattle bottom-up inventory Unless locally determined emissions factors for grazing cattle and feedlots are presented, use the emission factors recommended by the Australian Government (NIR 2017), alternatively provide extensively documented justification for using Harper et al. (1999) emission factors.

Closing Comments

The authors of Luhar et al. (2020) have an opportunity to update the inventory used as a prior for their Bayesian modelling and provide a transparent workflow that can be a template for other regions, both within Australia and worldwide.

There is a plethora of choices to be made when collating a regional bottom-up inventory, especially for any region with extensive gas production and agricultural activities. As currently documented in Luhar et al. (2020) the inventory cannot be validated. Thus, the prior used for the inverse Bayesian modelling cannot be validated. This distracts from the overall quality of the science that has been presented in other sections of Luhar et al. (2020), which comprehensively demonstrates the extent of coverage that can be obtained from just two greenhouse gas monitoring stations and highlights the enormous potential of similar setups for quantifying regional greenhouse gas fluxes throughout Australia.

The bottom-up inventory grazing cattle emissions may have been overestimated, and coal seam gas emissions appear to be underestimated. A number of other sources appear to have been overlooked. Redistributing the methane emissions to the correct sources and locations should improve the prior and the regional Bayesian inverse model estimates of methane emissions in the Surat Basin.

Clarity on how the bottom-up inventory emissions were estimated would greatly enhance the science outcomes reported in Luhar et al. (2020).

Regards

Bryce Kelly, PhD Associate Professor School of Biological, Earth and Environmental Sciences UNSW Sydney, 2052, NSW Australia

References

NIR (2017) National Inventory Report, 2015, Volume 1. Australian Government, Department of Industry, Science, Energy and Resources. The Australian Government Submission to the United Nations Framework Convention on Climate Change, Australian National Greenhouse Accounts, May 2017. https://publications.industry.gov.au/publications/climate-change/system/files/resources/gas-group/national-inventory-report-2015-vol1.pdf

NIR (2020) National Inventory Report, 2018, Volume 1. Australian Government, Department of Industry, Science, Energy and Resources. The Australian Government Submission to the United Nations Framework Convention on Climate Change, Australian National Greenhouse Accounts, May 2020. https://www.industry.gov.au/sites/default/files/2020-05/nga-national-inventory-report-2018-volume-1.pdf

API 2009 Compendium of Greenhouse Gas Emissions Methodologies For the Oil and Natural Gas Industry. https://www.api.org/~/media/Files/EHS/climate-change/2009_GHG_COMPENDIUM.pdf

Harper, L. A, Denmead, O. T., Freney, J. R, and Byers, F. M.: Direct measurements of methane emissions from grazing and feedlot cattle, Journal of Animal Science, 77(6), 1392–1401, https://doi.org/10.2527/1999.7761392x, 1999.

Iverach, C. P., Cendon, D. I., Hankin, S. I., Lowry, D., Fisher, R. E., France, J. L., Nisbet, E.G, Baker, A., Kelly, B. F. J. (2015). Assessing Connectivity Between an Overlying Aquifer and a Coal Seam Gas Resource Using Methane Isotopes, Dissolved Organic

[Figure]

Interactive
comment

Carbon and Tritium.. Scientific Reports, 5, 15996. doi:10.1038/srep15996

Kelly et al. (in preparation) What are bottom-up methane inventories missing? Insights from field measurements in coal seam gas fields, agricultural, and urban districts.

Lu, X., Harris, S. J., Fisher, R. E., Lowry, D., France, J. L., Hacker, J., Neininger, B., Röckmann, T., van der Veen, C., Menoud, M., Schwietzke, S., and Kelly, B. F. J. (2020) Methane Source Attribution Challenges in the Surat Basin, Australia, EGU General Assembly 2020, Online, 4–8 May 2020, EGU2020-12508, https://doi.org/10.5194/egusphere-egu2020-12508

Luhar, A., Etheridge, D., Loh, Z., Noonan, N., Spencer, D., Day, S. 2018. Characterisation of Regional Fluxes of Methane in the Surat Basin, Queensland. Final report on Task 3: Broad scale application of methane detection, and Task 4: Methane emissions enhanced modelling. Report to the Gas Industry Social and Environmental Research Alliance (GISERA). Report No. EP185211, October 2018. CSIRO Australia. https://publications.csiro.au/rpr/pub?pid=csiro:EP185211

Neininger, B., Hacker, J. M., and Lieff, W. (2020) Estimating Methane Emissions in the Surat Basin, Australia, including turbulent vertical Fluxes, EGU General Assembly 2020, Online, 4–8 May 2020, EGU2020-10993, https://doi.org/10.5194/egusphere-egu2020-10993

Nisbet, E. G., Fisher, R. E., Lowry, D., France, J. L., Allen, G., Bakkaloglu, S., Broderick, T.J., Cain, M., Coleman, M., Fermandez, J., Foster, G., Griffiths, P.T., Iverach, C.P., Kelly, B.F.J., Manning, M.R., Nisbet-Jones, P.B.R., Pyle, J.A., Townsend-Small, A., al-Shalaan, A., Warwick, N., Zazzeri, G. (2020). Methane Mitigation: Methods to Reduce Emissions, on the Path to the Paris Agreement. Reviews of Geophysics, 58(1). doi:10.1029/2019RG000675

Please also note the supplement to this comment:
https://www.atmos-chem-phys-discuss.net/acp-2020-337/acp-2020-337-SC1-

supplement.pdf

---

## Referee Comment (RC1) · Anonymous Referee #1 · 25 Jun 2020

This paper employs a dataset of quasi-continuous measurements over an 18-month period from two monitoring stations in the middle of a region characterized by a mix of largely anthropogenic methane sources to optimize gridded methane emission inventory estimates. It aims to scale inventory emission estimates for individual grid boxes with a focus on the coal seam gas (CSG) industry. Given the current lack of atmospheric data to inform CSG methane emissions in Australia and elsewhere, this paper is a useful addition to the literature to help researchers improve their methods to quantify emissions from this source. The analysis is very detailed, the paper is well written,

and the tables and figures are well presented.

However, I have two major comments/questions that may be important for the bottom-line implications of the study:

1. The background methane mole fraction estimation (Supplementary S3) requires some more discussion. As Figure 3 shows, both monitoring stations are surrounded by known methane sources that are being quantified here. The monitoring stations do not measure the background air entering the spatial domain for which the emissions are being quantified here (hence background estimation). Filtering peaks during the early afternoon may exclude the largest point sources, but not necessarily the area sources that are clearly shown to exist in Figure 3. Does this estimation method create a high bias for the background levels, and in extension a low bias for the posterior emissions (especially from distributed sources like CSG wells)? Could this explain why all inverse setups produce smaller posterior total emissions than the prior despite the acknowledgment in the paper that the inventory may miss some sources (so the inventory itself may be underestimated)? Note that the opposite is true when looking only at the CSG sub-domain, which is situated largely between both monitoring stations (thus the sources in the CSG sub-domain affect estimated background values to a lesser extent), which appears to underscore this conundrum. It is also noteworthy that such underestimation may be masked also in the q-q plots comparing observed and modeled concentrations because a potentially underestimated prior and overestimated background would compensate each other.

2. How are the higher-end modelled methane concentrations (but low occurrence, potentially not due to the infrequent emission, but rather due to their being point sources with fewer opportunities to be sampled) weighted against the overall average methane (but high occurrence) in the inversion model framework? Is this objectively weighted in the model (and if so, how), or is it a model design choice?

Below is a list of detailed comments that may help clarify arguments and language,

and correct potential errors.

Main article:

1.    Ln 39:   For balance, there's an ongoing discussion about the contrasting evidence (contemporary local measurements vs.    ice-core 14C data) regarding the magnitude of the fraction of natural geologic seepage: https://www.elementascience.org/articles/10.1525/elementa.383/

2.  Ln 58: "independent": I suggest "atmospherically based" instead since inverse estimates are by definition not completely independent of the prior/inventory.

3.  Ln 71: Through this or any top-down approach?  Would be valuable to mention if other top-down approaches have been used in Australia in the past.

4.  Ln 158: Would re-phrase that the two operators account for 1.5% of CSG production activity in the region, not emissions (which would be difficult to establish with any accuracy).

5.  Ln 189ff: Spatial resolution of $2.5° \times 2.5°$ means (roughly) 250 x 250 km2. How, then, is it possible to apply it at 5 x 5 km2?  Regarding the meaning of the 6 hour availability of met re-analyses, does it means that the temporal resolution is 6 hours?

6.  Ln 291: I assume you're referring to the bottom-up emission inventory?

7.  Ln 666: Arguably Figure 14b cannot be used to support the trend in the CSG activity data.  According to Ln 609, only 4% of the sub-domain emissions are due to CSG wells (and unclear whether the same processing facilities would emit more given more throughput), so any increase in well count may hardly be detectable by the monitoring stations. Thus, the insight here seems to be not that measurements aren't supporting the CSG increase, but that the existing monitoring setup is likely unable to detect.

Supplementary:

1. Ln 99: Emissions of methane due to incomplete combustion of CSG

2. Ln 100ff: Why are methane GWPs used for methane emissions from incomplete combustion, fugitives, and coal extraction? It sounds like the underlying EFs are given in CO2e, which seems illogical.

---

## Referee Comment (RC2) · Anonymous Referee #2 · 27 Jul 2020

The manuscript presented by Luhar and co-workers presents an analysis of methane emissions from a region in Queensland, Australia, that contains a mix of different source processes of which coal seam gas prodution is the one mostly targeted and discussed in the study. Overall the study used valid and up to date methods. The manuscript is well structured and easy to follow. Quantifying uncertain methane emissions on the regional scale by in-situ observations and atmospheric inversion techniques is an important task supporting emission reductions and as such the study deserves publication. However, the authors should include some additional discussion

of how their results may be used in the future by gas companies and/or authorities. I recommend the manuscript for publication after a number of minor issues (as listed below) are addressed/clarified by the authors.

Minor comments

Page 1, Line 2: Why is the term 'efficient' used in the title? What is efficient about this inversion approach? Further explain or omit from title.

P2, L58: Given the involved uncertainties in transport and inverse modeling, 'verification' may be a too strong term. Validation is often the preferred terminology.

Figure 1: A zoom into the study region including the location of the observational sites would be useful. This would also help to understand any orographic features of the domain.

P4, L90: Were the inlets mounted on small towers or on rooftops? Please briefly mention even if described elsewhere.

Bottom-up inventory: Which emission processes were separated for the agricultural sources? Enteric fermentation, manure handling, etc.? The information in the supplement is very brief and I was not able to obtain the cited report by Katestone. Since this is the dominating emission source in the area, it would be good to give a few more details and also to briefly discuss the uncertainties in these estimates.

P6, L145: What was the number of cattle in the feedlots? How do the emission factors per livestock unit compare between feedlots and free range? How were emissions from animal waste treated in the two cases?

P7, L164f: What is this rough estimate based on? It seems to be rather large considering that the main source is cattle and per livestock emission factors are more certain than 50 %. Is the livestock number that uncertain?

Figure 3: What is the reasoning about showing these specific towns? Is there any

larger population in the area?

Section 4.3: The analysis in the supplement is quite useful. How dose the wind rose comparison look for the filtered observation data. Does it improve? What is the mean bias for the filtered data? Next to wind speeds, mixing layer heights are critical when doing regional scale transport modeling and emission inversions. How is the mixing layer height treated in TAPM? Is there any way of comparing mixing layer heights for the target area and period or are their previous evaluations available for the model?

P12, L249f: Another important source of uncertainty is that of representativeness of the point measurement for the model grid cell (5x5 km). What are the observations compared to? Simulated values interpolated to the location of observation or grid cell containing the observation site? Are there any important sources in the closer vicinity of the sites (<10 km)?

P13, L282: Not immediately clear what top 5 % refers to. How do these top 5 % simulated events compare to the observations? Are these also the highest observed concentrations?

P19, L405ff: So if I understand this correctly, the source receptor relationship for a time t is constructed from output of c* at different times according to the value of tr at individual grid points. First, I am wondering if this could be illustrated for an example case where one would show the field c* for a given time and then the reconstructed source receptor relationship for the same time. Second, it seems that there will remain some form of smearing out of the transport history in time. How much does this conflict with filtering data by time of day instead of using the complete data set. Also what was the rational of using hourly data in this case instead of working with longer aggregation times for which the effect should be smaller?

P20, L436: What about the sub-grid variability of these sources? Is it kept for the transport simulation and a factor for the larger grid boxes optimised or is the emission flux constant within the large grid boxes. What about the different source categories?

Are they treated separately as was done for the forward simulation? Not clear from this description, later on it becomes clear that only total emissions are optimised.

P21, L460: Does high probability mean small uncertainty of the posterior? That would be surprising when starting from larger prior uncertainties.

P21, L461: Above, it was speculated that the uncertainty of the bottom-up approach was 50 %. Here it is suggested that 0.5 % should be used in an inversion. That seems to be a contradiction. Please elaborate on the small sigma_p. Also, is sigma_p the uncertainty of the total emissions in the inversion grid or that of individual grid cells?

Section 6.1: Usually, one would add random or auto-correlated noise to the synthetic observations as a test up to which degree of uncertainty the inversion can obtain useful information. Was this not done here at all?

Page 21, L464: How would the results change if only the synthetic observations were used at times when valid (filtered) observations were actually available? The latter was a considerably larger number of observations, so it is not clear how the results presented in this section can be propagated to the inversion with the more limited data set.

Section 6.2: Next to the posterior emissions it would be good to show simulated time series (synthetic obs, prior, posterior) and some performance stats in order to get a feeling for the inversion performance. This is done later on with additional forward simulations, but it should also be done with the concentrations directly obtained from the source receptor relationships and the coarse resolution emission setup as used in the inversion. Something to add to the supplement.

P22, L491: I don't like the terminology "no prior". There is a prior! Why not call the case "uniform" prior, which would describe the used PDF.

Section 7.1.3: Again it would be useful to see simulation performance for the three uncertainty levels.

P23, L535: So if the best estimate results from using a Gaussian prior distribution, I wonder why an MCMC approach was used at all. Wouldn't it be much more efficient to use the analytical solution of the Bayesian theorem for Gaussian PDFs in this case?

Figure 11b: Why not show the relative posterior uncertainty? Couldn't this be more directly compared to sigma_p?

P25, L551: Not clear which grid point this is referring to. Why is it relevant?

P26, L577: Give information on which case 3 inversion is used here (sigma_p=?).

P28, L611ff: This argument could also be supported by comparing the emissions from the non-CSG sub-domain. Do they differ significantly between bottom-up and posterior? If so, what are the possible reasons?

P29, L629: Which sigma_p level?

Figure 14: Include uncertainties. That would allow judging of how well 3-monthly emissions are constraint and if there is a real difference with time. Other studies have shown seasonality in agricultural emissions. Could this be a possibility here as well? Or does it have to do with a seasonality in the source receptor relationships?

Technical comments

P1,L21: 'identical TO' ...

Figure 5: Add explanation of dashed line to figure caption.

Figure 8: It seems to be more logical to start with the bottom-up emissions on the left (8a) and show the posterior on the right (8b).
* * *

---

## Author Comment (AC3) · 7 Oct 2020

We thank Assoc. Prof. Bryce Kelly for taking the time to read our manuscript and making a number of comments. We appreciate that he found our work 'a valuable scientific contribution.'

We are aware of the measurements and analysis work Prof. Kelly and his team have been doing on methane emissions from the Surat Basin, and it is really useful to get his perspective on our paper.

[Figure]
* * *
Interactive
comment

Virtually all comments by Prof. Kelly concern the construction of the bottom-up methane inventory for the Surat Basin that we have used in the paper.

The 2015 emission inventory for the Surat Basin was prepared by Katestone Environmental Pty. Ltd for us (i.e. CSIRO) (Lisa Smith of Katestone is a co-author on the paper).

We now attach the Katestone report "Surat Basin Methane Inventory 2015 – Summary Report" in the Supplement S6. This report fully explains as to how the bottom-up methane inventory was constructed and should answer many of the questions and comments by Prof. Kelly.

We want to emphasise that the main purpose of the paper was the top-down calculation (i.e. inverse Bayesian modelling). For this purpose, the bottom-up methane inventory provided a valuable a priori information. The inverse modelling aims to bring out any significant differences between the top-down and bottom-up estimates for methane emissions by using the long-term, in-situ methane concentration observations that we made at two locations, Ironbark and Burncluith, in the Surat Basin.

Our inverse modelling suggests the top-down emissions are 33% larger in the coal seam gas (CSG) subdomain and 18.5% lower in the non-CSG region (dominated by grazing cattle), than those from the bottom-up inventory. This result is qualitatively consistent with the calculations done by Prof. Kelly which give higher CSG emissions and lower grazing cattle emissions that our bottom-up inventory. We expand Section 7.3 of the paper to include this.

The 2015 bottom-up inventory was prepared by compiling the best information available at the time, and it was not in the scope of the work to go back and reconstruct the bottom-up inventory based on the results from the inverse modelling by re-examining the various source components.

Additional response to Prof. Kelly's is given below.

Comment: "As recently presented at EGU 2020 in Lu et al. (2020) UNSW researchers have developed their own bottom-up inventory in the Surat Basin for the year 2018. . . ."

Response: We are pleased to hear that Kelly et al. have prepared a bottom-up methane inventory for the Surat Basin for the year 2018 which they will shortly submit for review. Note that our inventory was for the year 2015, which partly corresponds the period of methane concentration measurements in the region (i.e. July 2015 – December 2016) used in our paper. We would expect changes in emissions going from 2015 to 2018, and would be keen in following up on Kelly et al.'s bottom-up inventory methodology when it is out in the print.

Comment: "Concerns with the lack of details provided on the inventory calculations. . ."

Response: As mentioned above, we now attach the Katestone report "Surat Basin Methane Inventory 2015 – Summary Report" in the Supplement S6, which provides full details of the bottom-up methodology adopted and data used.

Comment: "Suggested manuscript revision inventory calculations In the supporting information a table needs to be presented that lists the base quantity, emission factor used, clear referencing of the document(s) for the emission factor (and for each document clear referencing of the table and row selected for the emission factor), and justification for the selection of the emission factor, . . ."

Response: Again, most of this information is give in the Katestone report.

Comment: "Points of clarity required with the CSG bottom-up inventory estimation of emissions Because Luhar et al. (2020) does not adequately list the base CSG data . . ."

Response: Please see the Katestone report in the Supplement S6.

Comment: "No listing of Pels covered is provided in Luhar et al. (2020). From the Queensland Government database in the Surat Basin gas was produced from 3519 wells in the period ending 30/06/2015 and 3768 wells in the period ending 31/12/2015.

This total well number used in producing gas is actually slightly lower . . ."

Response: The total well number is lower probably because the database file places the gas fields of Spring Gully and Peat within the Bowen Basin whereas in our bottom-inventory these are part of the Surat Basin. This is because of how the gas field zones and basin boundaries are defined. The gas fields included in our study are based on their geographic locations relative to the square study domain selected. There is a footnote about this in Section 7.4 of the paper.

Comment: "It is well documented that there are emissions from the water management ponds in the Surat Basin: refer to Iverach et al. (2015) Figure 3, and Nisbet et al. (2020) Figure 10. . ."

Response: Methane emissions from produced water from both CSG production and processing are included in our emission inventory. These relate to collection and storage of produced water, and high point vents on produced water pipelines (Table 11 of the Katestone report). These emissions are calculated at $1.63 \times 10^6$ kg yr-1 ( 10% of the total CSG emissions). Produced water is a component of both CSG production and processing, and as such was not presented as a separate source category in the inventory but included under venting (now mentioned in the revised paper and Supplement S2.3).

We used an emission factor of 0.036 tonnes CH4/1000 m3 produced water based on API (2009) "Methane Emission Factors from Produced Water from Shallow Gas Wells" (76 psi ( 5.2 atmosphere) pressure or less and produced water at a temperature of $50°C$) (see API (2009) Table 5-11, page 5-57).

The above emission estimate from produced water is only about 10% of Prof. Kelly's estimate, which uses an emission factor of 0.31955 tonnes CH4/1000 m3 produced water, which is about 9 times greater that the value we used. This explains the discrepancy between the two estimates.

The emission factor Prof. Kelly used is also based on API (2009), but for "Produced Salt Water Tank Methane Flashing Emission Factors" (see API (2009) Table 5-10, page 5-57) with a separator pressure of 1000 psi ( 68 standard atmosphere pressure) and a produced salt water content of 10.7%. In this Table, the emission factor value varies greatly, from 1/35th to 1.25 times the value used by Prof. Kelly, depending on the separator pressure and produced water salt content. We are not sure how the emission factors given in Table 5-10 are applicable to the Surat Basin produced water.

Prof. Kelly does a bulk calculation for the total CSG emissions from the Surat Basin by using the total gas produced and Facility Level Average emission factors. The aim of our methodology was to develop a fine resolution (1 km $\times$ 1 km) gridded bottom-up inventory needed for forward modelling and inversion, by using information such as location of CSG well and processing facilities based on data available through DNRM, and methane emissions data and calculations provided by operators. There are differences between the two estimates, and as mentioned above, the inversion highlighted the differences between the top-down and bottom up estimates.

Also, please note that the Surat Basin as defined for the present study is the 350 km $\times$ 350 km area shown in Figure 1 (which is on Queensland's side) and does not cover the whole Surat Basin.

Reference: API 2009 Compendium of Greenhouse Gas Emissions Methodologies for the Oil and Natural Gas Industry. See https://www.api.org/~/media/files/ehs/climate-change/2009_ghg_compendium.ashx.

Comment: "Suggested manuscript revision CSG bottom-up inventory In the revised manuscript it is recommended for the CSG inventory portion ..."

Response: Again, much of the detail on our inventory emissions from the Surat Basin is given in the Katestone report, which is now included in the Supplement S6.

Comment: "Points of clarity required with the cattle bottom-up inventory estimation of

emissions The choice of using Harper et al. (1999) for cattle emission factors needs to be justified. . ."

Response: Please see the attached Katestone report.

We used the Harper et al. (1999) emission factor for cattle.

We acknowledge that it may be an overestimate, and if that is the case then our top-down emission estimate for the non-CSG area (dominated by grazing cattle) is consistent with this. We add the following new paragraph in Section 7.3

"Conversely, the total bottom-up inventory emissions from the non-CSG area is 125.5 $\times$ 10ˆ6 kg yr-1 whereas that obtained using the inversion (Case 3c) is 102.2 $\times$ 10ˆ6 kg yr-1 which is 18.5% lower than the former. The total bottom-up emission for this area is dominated by grazing cattle (62.7%), followed by feedlots (24.8%) and coal mines (8.6%), which together account for 96.1% of the emissions from this area. It is possible that the emission factor of 84 kg CH4 animal-1 yr-1 for Australian grazing cattle (Harper et al., 1991) used in the bottom-up inventory (see the Supplement S6) is an overestimate (cf. 51 kg CH4 animal-1 yr-1 for beef cattle (pasture) used by the Australian National Inventory Report (NIR, 2017) or 63 kg CH4 animal-1 yr-1 for non-dairy cattle for the Oceania (IPCC, 2019)), and that would be consistent with the lower top-down methane emission from the non-CSG area compared to the inventory. This also means that the CSG component of the top-down emissions in CSG sub-domain could be higher to compensate for the lower grazing cattle emissions if a lower emission factor for grazing cattle is used."

"Closing Comments. . ."

Thanks again for your comments.

Please also note the supplement to this comment:
https://acp.copernicus.org/preprints/acp-2020-337/acp-2020-337-AC3-supplement.pdf

---

## Author Response (AR1)

**Reply by the authors to Referee #1's comments on**
**"Quantifying methane emissions from Queensland's coal seam gas producing Surat Basin using inventory data and an efficient regional Bayesian inversion" (#acp-2020-337)**

**Anonymous Referee #1 (RC1)**

We are grateful to the Referee for taking the time to read our manuscript and making a number of valuable comments. In the following, we provide our responses to these comments (the Referee's comments are shown in blue). The locations of the changes made refer to those in the non-tracked version of the revised manuscript.

This paper employs a dataset of quasi-continuous measurements over an 18-month period from two monitoring stations in the middle of a region characterized by a mix of largely anthropogenic methane sources to optimize gridded methane emission inventory estimates. It aims to scale inventory emission estimates for individual grid boxes with a focus on the coal seam gas (CSG) industry. Given the current lack of atmospheric data to inform CSG methane emissions in Australia and elsewhere, this paper is a useful addition to the literature to help researchers improve their methods to quantify emissions from this source. The analysis is very detailed, the paper is well written, and the tables and figures are well presented.

**Response:** Thank you for your comments.

However, I have two major comments/questions that may be important for the bottom-line implications of the study:

1. The background methane mole fraction estimation (Supplementary S3) requires some more discussion. As Figure 3 shows, both monitoring stations are surrounded by known methane sources that are being quantified here. The monitoring stations do not measure the background air entering the spatial domain for which the emissions are being quantified here (hence background estimation). Filtering peaks during the early afternoon may exclude the largest point sources, but not necessarily the area sources that are clearly shown to exist in Figure 3. Does this estimation method create a high bias for the background levels, and in extension a low bias for the posterior emissions (especially from distributed sources like CSG wells)? Could this explain why all inverse setups produce smaller posterior total emissions than the prior despite the acknowledgment in the paper that the inventory may miss some sources (so the inventory itself may be underestimated)? Note that the opposite is true when looking only at the CSG sub-domain, which is situated largely between both monitoring stations (thus the sources in the CSG sub-domain affect estimated background values to a lesser extent), which appears to underscore this conundrum. It is also noteworthy that such underestimation may be masked also in the q-q plots comparing observed and modeled concentrations because a potentially underestimated prior and overestimated background would compensate each other.

**Response:** The reviewer has a valid point. Specification of background in a regional model is tricky. Ideally, this requires methane measurements at many locations around the perimeter of the study domain or modelling methane at much larger scale (preferably global), with all sources, sinks and chemical processes accounted for, which could then provide concentration boundary conditions needed for the regional modelling. Notwithstanding the difficulty in carrying out such a major computational task, there are modelling difficulties and uncertainties associated with emissions, representation of processes, model resolution issues etc. There could be other ways to calculate

background too, such as satellite data and model-data assimilation. Nevertheless, we believe that for the hourly-averaged, ground-level background concentrations needed in regional modelling study like ours, in-situ observations near the ground are still a better means to derive the background provided there are sufficient number of monitors sited at favourable locations than using a larger scale model.

In our case, we are limited by only two monitors (i.e. Ironbark and Burncluith) within a relatively large study domain. This reflects the operational and budget constraints of this project and is likely typical of many others. We calculated the hourly background using a methodology described in the Supplement S3 that utilised methane concentration measurements from the two monitors. It assumes that under vigorous atmospheric mixing conditions in the daytime, the measured concentrations within study domain represent methane levels both within and outside the domain boundaries, so that the measured concentrations can be taken to represent the background under such conditions. Figure 4 in the paper shows how the derived background defines the baseline for the methane measurements, which we have treated as the real background.

Because the background concentration is calculated from the measurements within the source region under study, there is a possibility that it represents an upper limit on the magnitude of the background, meaning that the real background is potentially lower than what we have used (as alluded to by the referee).

To examine the sensitivity of the emission inference to the background methane, we have done an additional inversion using an alternate background time series and this is described in detail in the new Supplement S5. The alternate background was constructed using our original background methane and marine baseline methane measurements from the Cape Grim Baseline Air Pollution Station (https://capegrim.csiro.au), located on the north-west tip of Tasmania (40.7ºS, 144.7ºE) (see the Supplement S5). The measurements from the Station were filtered for the marine baseline air (in southern mid latitudes), and the baseline methane thus represents concentration levels without the direct influence of the continental sources. As shown in Figure 1 below, the alternate background falls between the Surat Basin background as used in our study and the Cape Grim baseline (i.e. between the two bounds), and is, on average, lower than the previously used Surat background by 2.8 ppb. (On average, the Cape Grim marine baseline was 8.4 ppb lower than the original Surat background used).

[Figure]

**Figure 1.** The average hourly background $CH_4$ concentration (ppbv) time series (green line) as used in the present paper. The hourly-averaged Cape Grim baseline methane is shown as a red line. Blue line is the alternate background.

The inversion results in Table 1 below show that compared to the inferred emissions obtained using the original background methane the alternate background gives total emissions that are 6.8% higher, while the increase is smaller at 3.9% in the CSG subdomain and larger at 8.5% in the non-CSG region. The overall increase is expected because the increase in the measured concentrations by 2.8 ppb as a result of the use of the alternate background needs to be accounted for by the inversion by enhancing the amount of inferred emissions.

We also find that the amount of increase in the inferred emissions with the alternate background is almost uniformly spread through the study domain relative to the total emission, and that there are no significant spatial distributional shifts in the inferred emissions with the two background choices. This means that if these enhanced emissions are used in a forward model simulation, they would lift the modelled concentrations throughout the region by a very similar amount (likely by 2.8 ppb).

**Table 1: Inferred emissions ($\times 10^6$ kg yr$^{-1}$) obtained using the original methane background variation used in the paper (Case 3c in the paper, with the bottom-up inventory as a Gaussian prior with 3% uncertainty relative to the mean) and those obtained using the alternate methane background variation. The values in the parentheses are % change over the original inferred emissions.**

| Methane background | Total | CSG subdomain | Non-CSG subdomain |
|---|---|---|---|
| Original background (as used in the paper) | 165.8 | 63.6 | 102.2 |
| Alternate background | 177.0 (+6.8%) | 66.1 (+3.9%) | 110.9 (+8.5%) |

The above analysis demonstrates that there is an increase in the amount of inferred emissions with the alternate background and that this increase is smaller in the CSG subdomain relative to the original inferred emission.

**Changes in manuscript:** The new Supplement S5 given with full details of the above calculation, and the results are also summarised in Section 7.5 of the revised paper.

2. How are the higher-end modelled methane concentrations (but low occurrence, potentially not due to the infrequent emission, but rather due to their being point sources with fewer opportunities to be sampled) weighted against the overall average methane (but high occurrence) in the inversion model framework? Is this objectively weighted in the model (and if so, how), or is it a model design choice?

**Response:** In our inversions, the hourly-averaged methane measurements obtained during July 2015–December 2016 are combined in one Bayesian calculation to derive time invariant top-down emissions on an $11 \times 11$ source grid within the domain. Our inverse model framework is, in principle, able to discriminate between a source with a high emission rate but with infrequent impact at a sampling point and a source with a low emission rate but with frequent impact at the sampling point. This is because the concentration observations at the sampling point would reflect representative signals from these two types of sources, and this information when used in the source-receptor relationship would optimise the source emission rates accordingly such that they

best describe the concentration observations. In practice, however, the success in discriminating sources depends on the quality and quantity of available concentration observations, their spatial coverage, and on the number of source parameters that need to be quantified. This is where the specification of the prior plays a very important role because the information available (through concentration observations) may not be adequate to estimate the source parameters properly. This is demonstrated in our study.

Therefore, essentially, the only source weighting in our inverse framework is through the specification of the prior, and there is no other source weighting included/needed in the model apart from what is implicit through the Bayesian approach.

**Changes in manuscript:** We do not think that there is any change needed in the paper and hope that the above clarification is satisfactory.

Below is a list of detailed comments that may help clarify arguments and language, and correct potential errors.

Main article:

1. Ln 39: For balance, there's an ongoing discussion about the contrasting evidence (contemporary local measurements vs. ice-core 14C data) regarding the magnitude of the fraction of natural geologic seepage: https://www.elementascience.org/articles/10.1525/elementa.383/

**Response:** We have included two references to the bottom-up global estimates of natural geologic seepage.

**Changes in manuscript:**

We have modified the original wording to:

"However, a study using measurements of carbon-14 in methane recently showed that nearly all methane from fossil sources is anthropogenic, contrasting with the bottom-up estimates of significant natural geologic seepage (Etiope et al., 2019; Etiope and Schwietze, 2019), and that fossil fuel methane emissions may be underestimated by up to 40% (Hmiel et al., 2020)."

References:

Etiope, G, Ciotoli, G, Schwietzke, S and Schoell, M. 2019. Gridded maps of geological methane emissions and their isotopic signature. Earth Syst Sci Data 11: 1–22. DOI: 10.5194/essd-11-1-2019

Etiope, G. and Schwietzke, S., 2019. Global geological methane emissions: an update of top-down and bottom-up estimates. Elem Sci Anth, 7(1), p.47. DOI: http://doi.org/10.1525/elementa.383

2. Ln 58: "independent": I suggest "atmospherically based" instead since inverse estimates are by definition not completely independent of the prior/inventory.

**Response:** Point taken.

**Changes in manuscript:** Modification made.

3. Ln 71: Through this or any top-down approach? Would be valuable to mention if other top-down approaches have been used in Australia in the past.

**Response:** Point taken.

**Changes in manuscript:** We have added the following text:

"To our knowledge, this study is the first in Australia to quantify regional scale $CH_4$ emissions through a top-down approach employing transport modelling and concentration measurements, although studies at other spatial scales with broadly similar approaches have been reported, e.g. by Luhar et al. (2014) and Feitz et al. (2018) for single point sources at local scale and by Wang and Bentley (2002) at continental scale with Australian methane emissions divided into eight source regions."

References:

Luhar et al. (2014) and Feitz et al. (2018) already cited in the paper.

Wang, Y. P., and S. T. Bentley, S. T.: Development of a spatially explicit inventory of methane emissions from Australia and its verification using atmospheric concentration data, Atmospheric Environment, 36, 4965–4975, https://doi.org/10.1016/S1352-2310(02)00589-7, 2002.

4. Ln 158: Would re-phrase that the two operators account for 1.5% of CSG production activity in the region, not emissions (which would be difficult to establish with any accuracy).

**Response:** Point taken.

**Changes in manuscript:** The sentence is changed to "…but it was established that these two operators, with a total of 256 wells, only accounted for about 1.5% of the CSG activities that may be related to emissions."

5. Ln 189ff: Spatial resolution of 2.5º × 2.5º means (roughly) 250 x 250 km2. How, then, is it possible to apply it at 5 x 5 km2? Regarding the meaning of the 6 hour availability of met re-analyses, does it means that the temporal resolution is 6 hours?

**Response:** There is some misunderstanding here. The spatial resolution of 2.5º × 2.5º corresponds to the synoptic-scale fields of the horizontal wind components, temperature and moisture that are required as input boundary conditions for the outermost domain of TAPM. These fields given at 6-hourly intervals were sourced from the U.S. NCEP (National Centers for Environmental Prediction) reanalysis database. The TAPM model outputs hourly-averaged fields of meteorology and concentration at a specified horizontal resolution, which in the present application was 5 km × 5km.

**Changes in manuscript:** The above has been made clearer in the 2nd last paragraph of Section 4.1 of the revised paper (lines 212-220). Some more details of the model are given in the 2nd paragraph of this Section (lines 193-204).

6. Ln 291: I assume you're referring to the bottom-up emission inventory?

**Response:** Yes. Thanks for pointing that out. Correction made.

**Changes in manuscript:** As above.

7. Ln 666: Arguably Figure 14b cannot be used to support the trend in the CSG activity data. According to Ln 609, only 4% of the sub-domain emissions are due to CSG wells (and unclear whether the same processing facilities would emit more given more throughput), so any increase in well count may hardly be detectable by the monitoring stations. Thus, the insight here seems to be not that measurements aren't supporting the CSG increase, but that the existing monitoring setup is likely unable to detect.

**Response:** We have modified the text to improve clarity. A curve for the number of wells is also included in Figure 16 (Figure 19 in the revised paper).

**Changes in manuscript:** The paragraph revised as follows (lines 810-818):

"However, Figure 19 (*which is old Figure 16*) also shows that there is a downward trend in the amount of flared/vented gas. Considering, based on the bottom-up inventory in Section 3, that venting (from processing) is the biggest contributor (88%) followed by flaring (8%) (from both processing and production) to the total CSG methane emissions, it is plausible that despite the increase in the CSG development in the area the CSG-related methane emissions have not increased, and that they may have even gone down. The temporal variation of the inferred emissions in Figure 17b (*which is old Figure 14b*) for the CSG dominated area also does not indicate any consistent increase in emissions from 2015 to 2016. Thus, the 33% higher top-down emission estimate from the CSG area compared to the inventory estimate cannot be explained in terms of the growth in the CSG production from 2015 to 2016 and is possibly related to underestimated or missing emissions in the inventory. This also implies that the emissions from CSG may be more closely related to practices in the industry than to the amount of CSG produced."

Supplementary:

1. Ln 99: Emissions of methane due to incomplete combustion of CSG

**Response:** Change made.

**Changes in manuscript:** As above.

2. Ln 100ff: Why are methane GWPs used for methane emissions from incomplete combustion, fugitives, and coal extraction? It sounds like the underlying EFs are given in CO2e, which seems illogical.

**Response:** The calculation methods used to estimate methane emissions from CSG activities are consistent with the Australian National Greenhouse and Energy Reporting (NGER) program. We now attach the Katestone report "*Surat Basin Methane Inventory 2015 – Summary Report*" in the Supplement S6 of the paper, which explains in full detail how these emissions were calculated.

**Changes in manuscript:** As above.

**Reply by the authors to Referee #2's comments on**
**"Quantifying methane emissions from Queensland's coal seam gas producing Surat Basin using inventory data and an efficient regional Bayesian inversion" (#acp-2020-337)**

**Anonymous Referee #2 (RC2)**

We are grateful to the Referee for taking the time to read our manuscript and making a number of valuable comments. In the following, we provide a response to these comments (the Referee's comments are shown in blue). The locations of the changes made refer to those in the non-tracked version of the revised manuscript.

The manuscript presented by Luhar and co-workers presents an analysis of methane emissions from a region in Queensland, Australia, that contains a mix of different source processes of which coal seam gas production is the one mostly targeted and discussed in the study. Overall the study used valid and up to date methods. The manuscript is well structured and easy to follow. Quantifying uncertain methane emissions on the regional scale by in-situ observations and atmospheric inversion techniques is an important task supporting emission reductions and as such the study deserves publication. However, the authors should include some additional discussion of how their results may be used in the future by gas companies and/or authorities. I recommend the manuscript for publication after a number of minor issues (as listed below) are addressed/clarified by the authors.

**Response:** Thank you for your comments.

**Changes in manuscript:** Regarding some additional discussion of how the results may be used in the future by gas companies and/or authorities, we have included the following text at the end in Conclusions (lines 904-913):

"The methods developed in this study could be used to improve the monitoring and management of greenhouse gas and other air emissions from the onshore gas industry, including that in the Surat Basin. They provide independent information to industry and communities living in gas development regions on one of the main environmental impacts potentially arising from onshore gas developments. Improved quantification of methane emissions on the regional scale is an important step in emissions reductions from the onshore gas sector and possibly other industries. The present top-down method is particularly suited to distributed emissions with potentially unknown locations across a large geological gas reservoir and gas production infrastructure. If monitoring is deployed before gas exploration and production begins then a baseline would be established from which emissions from the industry might be detected. Ongoing top-down quantification, with monitoring stations located close to where emissions appear and with source-specific information from tracers could provide the information necessary to validate emissions from the gas industry to support greenhouse gas inventories."

Minor comments

Page 1, Line 2: Why is the term 'efficient' used in the title? What is efficient about this inversion approach? Further explain or omit from title.

**Response:** We have decided to omit the term 'efficient' from the title. The reason for its use was the application of the MCMC sampling method and the backward plume approach which make

computations very efficient. However, we admit that this is not the first time these approaches have been used in inverse modelling in general.

**Changes in manuscript:** 'efficient' omitted in the title.

P2, L58: Given the involved uncertainties in transport and inverse modeling, 'verification' may be a too strong term. Validation is often the preferred terminology.

**Response:** Point taken. 'Verification' replaced by 'validation'.

**Changes in manuscript:** As above.

Figure 1: A zoom into the study region including the location of the observational sites would be useful. This would also help to understand any orographic features of the domain.

**Response:** Point taken. We include an orographic map (Figure 1b) and also a Google Earth map showing the surface characteristics (Figure 1c) of the study domain. The Ironbark and Burncluith monitoring sites and the three biggest towns in the area are also shown.

**Changes in manuscript:** As above.

P4, L90: Were the inlets mounted on small towers or on rooftops? Please briefly mention even if described elsewhere.

**Response:** Inlets were mounted on masts.

**Changes in manuscript:** At line 103, we say '…with inlets placed on masts at a height of 10 m'.

Bottom-up inventory: Which emission processes were separated for the agricultural sources? Enteric fermentation, manure handling, etc.? The information in the supplement is very brief and I was not able to obtain the cited report by Katestone. Since this is the dominating emission source in the area, it would be good to give a few more details and also to briefly discuss the uncertainties in these estimates.

**Response:** We have now included the full Katestone report "*Surat Basin Methane Inventory 2015 – Summary Report*" in the Supplement S6 (it was prepared for us, i.e. CSIRO, by Katestone). It provides a comprehensive detail as to how the bottom-up inventory was constructed (largely by Lisa Smith of Katestone, who is a co-author on the present paper), including agricultural sources and uncertainties.

**Changes in manuscript:** As above.

P6, L145: What was the number of cattle in the feedlots? How do the emission factors per livestock unit compare between feedlots and free range? How were emissions from animal waste treated in the two cases?

**Response:** We now give the Katestone report in the Supplement S6 which provides this information.

**Changes in manuscript:** As above.

P7, L164f: What is this rough estimate based on? It seems to be rather large considering that the main source is cattle and per livestock emission factors are more certain than 50 %. Is the livestock number that uncertain?

**Response:** Yes, this was a very rough estimate, and we do not have any solid justification for it. Therefore, we have decided to delete it and modify the paragraph. The Katestone report that we now provide in the Supplement S6 provides more information about the bottom-up emissions.

**Changes in manuscript:** We have deleted this sentence.

Figure 3: What is the reasoning about showing these specific towns? Is there any larger population in the area?

**Response:** These are only given as reference points. We think that not all town locations are necessary. We now only present the locations of the three biggest towns, i.e. Dalby, Roma and Chinchilla (population 12700, 6850 and 6600, respectively), in the region.

**Changes in manuscript:** The above is stated in the Figure 1 and Figure 3 captions.

Section 4.3: The analysis in the supplement is quite useful. How does the wind rose comparison look for the filtered observation data. Does it improve? What is the mean bias for the filtered data? Next to wind speeds, mixing layer heights are critical when doing regional scale transport modeling and emission inversions. How is the mixing layer height treated in TAPM? Is there any way of comparing mixing layer heights for the target area and period or are their previous evaluations available for the model?

**Response:** In the Supplement S4, we now present a wind rose comparison for the filtered data (Figure S4) and provide the corresponding model performance statistics for meteorology (Table S1). With the filtering, the mean wind speed is predicted slightly worse, but the wind components are predicted better, which implies that there is an improvement in the estimation of wind direction with filtering.

The mean bias for the unfiltered and filtered data is now reported in Table S1.

Regarding mixing height, because TAPM is a fully prognostic, coupled meteorological and dispersion model, the predicted three-dimensional meteorological and turbulence fields are used directly by the dispersion component to predict concentrations. Therefore, there is no explicit use of mixing height as a parameter and the atmospheric mixing is taken care of by the predicted turbulence fields. Some of the model parameters that represent turbulence (and hence mixing) include friction velocity (mechanical turbulence) and surface heat flux (buoyancy-generated turbulence) have previously been evaluated in some of the studies cited (e.g. Luhar and Hurley, 2003; Hurley and Luhar, 2009; Luhar and Hurley, 2012; and Luhar et al., 2014)

In the Supplement S4, the link https://scholar.google.com.au/scholar?oi=bibs&hl=en&cites=13876071272134760358 to TAPM citation database provides additional references for TAPM application and evaluation.

**Changes in manuscript:** The Supplement S4 is modified with new Figure S4 and Table S1 included.

We provide some additional information about the meteorological component of the model in Section 4.1, which also details how turbulence is calculated (lines 193-204).

Modified paragraph:

"The model has previously been applied to a variety of flow, turbulence and dispersion problems at various scales, such as those reported by Luhar and Hurley (2003), Luhar et al. (2008), Hurley and Luhar (2009), Luhar and Hurley (2012), Luhar et al. (2014), Matthaios et al. (2017), and Luhar et al. (2020), which include model evaluation studies."

P12, L249f: Another important source of uncertainty is that of representativeness of the point measurement for the model grid cell (5x5 km). What are the observations compared to? Simulated values interpolated to the location of observation or grid cell containing the observation site? Are there any important sources in the closer vicinity of the sites (<10 km)?

**Response:** The hourly-averaged model predictions on the innermost grid domain were extracted at the lowest model level (10 m) at the grid point nearest to each of the monitoring sites for comparison with the observations. This is now stated in the text.

We agree that the model's representation of point measurements by grid-cell averaged values is another source of uncertainty, and it is now stated in the text.

The location of the two measurement stations was based on criteria given in Section 2, first paragraph, to "optimise the size and frequency of detection of methane emissions from the broader CSG source region without being unduly impacted by individual sources in the proximity of the measurement sites". (Other practical considerations are noted in the reference (Day et al., 2015), namely access, power, security, landowner assistance and possible future developments that would impact the site.) The sites were selected to avoid potential large, sustained methane sources within 10-20 km or even small sources within about a kilometre of the measurement inlet. Surveys of maps and by vehicle involving mobile methane monitoring of the area around the site identified few such sources. Small sources that were closer to the inlets (mainly Burncluith) were identified and their signals filtered from the data as described in Section 2. As a result, we expect that the hourly-averaged filtered data (Section 2) are as representative as possible of the atmospheric methane concentration across the $5 \times 5$ km grid cell containing the observation site, and can be directly compared to the model simulations.

**Changes in manuscript:** As above is summarise in the first para of Section 4.4, and it is mentioned that this is another possible source of differences between the observations and model predictions.

P13, L282: Not immediately clear what top 5 % refers to. How do these top 5 % simulated events compare to the observations? Are these also the highest observed concentrations?

**Response:** These are the highest 5% of the modelled concentrations, i.e. all the values above the 95th percentile. The idea here was to determine the dominant source types that contribute to highest modelled concentrations. A comparison of the modelled and observed concentrations has already been made in Figures 5 and 6, and it is clear that the highest concentrations are generally underestimated by the model at both sites (more so at Ironbark).

**Changes in manuscript:** The sentence is modified to '… the highest 5% of the modelled hourly-averaged methane concentrations (i.e. all the concentrations above the 95th percentile)'.

P19, L405ff: So if I understand this correctly, the source receptor relationship for a time t is constructed from output of c* at different times according to the value of tr at individual grid

points. First, I am wondering if this could be illustrated for an example case where one would show the field c* for a given time and then the reconstructed source receptor relationship for the same time. Second, it seems that there will remain some form of smearing out of the transport history in time. How much does this conflict with filtering data by time of day instead of using the complete data set. Also what was the rational of using hourly data in this case instead of working with longer aggregation times for which the effect should be smaller?

**Response:** The Referee is correct. We now explain it a bit better in the text and also present an illustrative example case in new Figure 7 that shows the field $c*$ for a given time and the reconstructed source receptor relationship for the same time (lines 440-461).

Occasionally, there may be some remains of smeared out transport history in time, but generally the intensity and the frequency of this is very small.

We do not think our method of reconstructing the hourly source-receptor relationship would conflict with the filtering of the data. This relationship is continuous with time, and its value at a particular hour would match the data points at that hour.

The rationale of using the hourly-averaged data rather (for which the effect of transport history would be smaller) was to maximise on the available information to constrain the inversions better. We could use longer averages, but that would have reduced the number of concentration data. Also, the wind direction variation inherent in the hourly data aids in better 'triangulation' of sources; the degree of this variation is progressively reduced as the averaging times are made longer. However, one could use longer aggregation times to see what difference that makes, but we have not attempted that. (Lines 493-497).

**Changes in manuscript:** As above, and new Figure 7 (a, b, c).

P20, L436: What about the sub-grid variability of these sources? Is it kept for the transport simulation and a factor for the larger grid boxes optimised or is the emission flux constant within the large grid boxes. What about the different source categories? Are they treated separately as was done for the forward simulation? Not clear from this description, later on it becomes clear that only total emissions are optimised.

**Response:** For the purposes of inferring emission rates using the inverse modelling, $11 \times 11$ source grid points are considered within the study domain. No sub-grid variability of these emission rates is considered. Given the limitation as to the type and amount of concentration observations we have for inversion, the inverse methodology used does not distinguish between different source categories. This is mainly because the concentration of methane alone was monitored and not tracers specific to methane source types. Therefore, there are no separate sources categories in the inferred emissions, unlike what was done for the forward simulation - only total emissions are optimised.

**Changes in manuscript:** This is clarified in the text (lines 500-507).

P21, L460: Does high probability mean small uncertainty of the posterior? That would be surprising when starting from larger prior uncertainties.

**Response:** The sentence is not correct and has been deleted.

**Changes in manuscript:** As above.

P21, L461: Above, it was speculated that the uncertainty of the bottom-up approach was 50 %. Here it is suggested that 0.5 % should be used in an inversion. That seems to be a contradiction. Please elaborate on the small sigma_p. Also, is sigma_p the uncertainty of the total emissions in the inversion grid or that of individual grid cells?

**Response:** We are not confident about the previously speculated uncertainty of 50% in the bottom-up approach, and have, therefore, deleted the sentence.

Following this comment and another comment below by the Referee, we revised Section 6 on inversion using the 'synthetic' concentration data considerably, with new model runs using an increased prior uncertainty (5% and 10%) and only considering times when the valid (or filtered) observations were actually available. This is more realistic, and the results now provide a better guidance to inversion using the real data.

sigma_p is the uncertainty (standard deviation) in the prior of the individual source and is specified as % relative to the prior mean value (there are $11 \times 11$ sources considered for the emission inference).

**Changes in manuscript:** Section 6 revised, with clarification in the text.

Section 6.1: Usually, one would add random or auto-correlated noise to the synthetic observations as a test up to which degree of uncertainty the inversion can obtain useful information. Was this not done here at all?

**Response:** We did not add any random or auto-correlated noise to the synthetic observations. But we performed new synthetic runs with the same 3.5 ppb uncertainty in the synthetic concentrations as that in the concentration observations for real inversions.

**Changes in manuscript:** The synthetic inversion Section 6 modified (lines 514-586) with modified and new plots (Figures 9–11).

Page 21, L464: How would the results change if only the synthetic observations were used at times when valid (filtered) observations were actually available? The latter was a considerably larger number of observations, so it is not clear how the results presented in this section can be propagated to the inversion with the more limited data set.

**Response:** This is a valid point. Following the Referee comment, we have revised the section on "Inversion using the 'synthetic' concentration data" considerably. We now present inversion results by using the synthetic observations only for times when the valid (or filtered) observations were actually available. The uncertainty in the synthetic concentrations is now taken to be the same (i.e. 3.5 ppb) as that in the concentration observations for real inversions. This now provides a better propagation of the results presented in this section to the next section on inversion using the real observations.

**Changes in manuscript:** The synthetic inversion Section 6 modified (lines 514-586) with modified and new plots (Figures 9–11).

Section 6.2: Next to the posterior emissions it would be good to show simulated time series (synthetic obs, prior, posterior) and some performance stats in order to get a feeling for the inversion performance. This is done later on with additional forward simulations, but it should also

be done with the concentrations directly obtained from the source receptor relationships and the coarse resolution emission setup as used in the inversion. Something to add to the supplement.

**Response:** Good point, but because this is a case of synthetic time series, we thought that rather than presenting the simulated time series it would be better to actually compare the inferred emissions with the bottom-up inventory emissions that were used to simulate the 'synthetic' concentrations (which in turn were used in the inversion). We have done this exercise and presented the results in Section 6.2 along with some performance statistics (i.e. linear least-squares fits and correlation coefficient). These new results also lead to a better linkage of this section on synthetic inversion to the next section on real inversion.

**Changes in manuscript:** As above (lines 535–586). New figures 10 and 11. Modified Figure 9.

P22, L491: I don't like the terminology "no prior". There is a prior! Why not call the case "uniform" prior, which would describe the used PDF.

**Response:** We now call it non-informative uniform prior in the text.

**Changes in manuscript:** As above.

Section 7.1.3: Again it would be useful to see simulation performance for the three uncertainty levels.

**Response:** Simulation performance for the three uncertainty levels is now given in new Table 1. We thought it was more appropriate to give it in Section 7.2 on validation than in Section 7.1.3.

**Changes in manuscript:** As above.

P23, L535: So if the best estimate results from using a Gaussian prior distribution, I wonder why an MCMC approach was used at all. Wouldn't it be much more efficient to use the analytical solution of the Bayesian theorem for Gaussian PDFs in this case?

**Response:** The Referee is correct with the Gaussian prior distribution. However, our idea was to formulate our inverse modelling tool with MCMC so that it is more generally applicable than just for the Gaussian PDFs—something that could be useful for future applications that we may consider.

**Changes in manuscript:** None.

Figure 11b: Why not show the relative posterior uncertainty? Couldn't this be more directly compared to sigma_p?

**Response:** Point taken. The plot has been replaced by the relative (%) posterior uncertainty and the corresponding text modified accordingly.

**Changes in manuscript:** As above.

P25, L551: Not clear which grid point this is referring to. Why is it relevant?

**Response:** This is grid point (11, 4), which corresponds to a relatively strong coal mine source in the bottom-up inventory (Figure 3d).

**Changes in manuscript:** Change made in the text.

P26, L577: Give information on which case 3 inversion is used here (sigma_p=?).

**Response:** This is Case 3 with 3% prior uncertainty relative to the mean.

**Changes in manuscript:** Change made in the text.

P28, L611ff: This argument could also be supported by comparing the emissions from the non-CSG sub-domain. Do they differ significantly between bottom-up and posterior? If so, what are the possible reasons?

**Response:** Emissions from the non-CSG subdomain are now compared (lines 739-742), new plot 17c and Table 2 and the discussion.

**Changes in manuscript:** As above.

**Response:** This is Case 3 with 3% prior uncertainty relative to the mean.

**Changes in manuscript:** Change made in the text.

Figure 14: Include uncertainties. That would allow judging of how well 3-monthly emissions are constraint and if there is a real difference with time. Other studies have shown seasonality in agricultural emissions. Could this be a possibility here as well? Or does it have to do with a seasonality in the source receptor relationships?

**Response:** Uncertainties are now included. There is also an additional plot (Figure 17c) for the 3-monthly variation of the inferred emissions for the non-CSG area (which is dominated by grazing cattle emissions as per the bottom-up inventory). In this plot, we also present a 3-monthly climatological average (1992 – current 2020) of rainfall at the Dalby airport, located next to the town of Dalby, within the study domain. There is a good correlation ($r = 0.79$) between the non-CSG area methane emissions and the rainfall, suggesting that the 3-monthly emission variation could possibly be explained in terms of the seasonality in agricultural and wetland emissions influenced by rainfall.

Another potential contributor to the temporal variability in the inferred emissions is the seasonality of the winds in the area which influence the source-receptor relationships. We have not explored this possibility here.

**Changes in manuscript:** As above. Figure 17c included. Lines 772-783.

Technical comments

**Response:** Correction made.

Figure 5: Add explanation of dashed line to figure caption.

**Response:** Point taken.

**Changes in manuscript:** We say '…and the dashed line is the 1:1 line (i.e. perfect agreement)'.

Figure 8: It seems to be more logical to start with the bottom-up emissions on the left (8a) and show the posterior on the right (8b).

**Response:** Point taken.

**Changes in manuscript:** The plots have been swapped and the figure caption and text modified accordingly.

[revised manuscript text omitted]

490    As an example, Figure 8a presents the modelled backward concentration field (/q, s m⁻³) due to a unit point release (q = 1 g s⁻¹) averaged over all hourly fields over the simulation period for Ironbark. Essentially, the value at any point in Figure 8a is equivalent to the simulation average forward model concentration value at this monitoring location if there were a source at that point with unit emission. Put differently, tThe backward concentration value at a given location represents the probability (including both frequency and intensity) a source emission at that location adds to the concentration at the monitoring site. The
495    backward field is mainly determined by flow the field across the domain and the separation between the receptor and the source.

[Figure]

500    **Figure 8. Normalised modelled backward distribution of near-surface concentration ($c^*/q$, × 10⁻⁹ s m⁻³), which is an average over the entire study period: (a) Ironbark, and (b) Burncluith.**

The modelled backward concentration field ($c^*/q$, s m⁻³) averaged over all hourly fields over the simulation period for Ironbark is shown in Figure 8a, which suggests that, It is apparent from Figure 8a that overall, any sources located farther from the
505    monitoring station would contribute less as plume concentrations decrease with increasing distances, and vice versa. The directional distribution of the backward field is also a function of the distribution of regional winds which determine how often the receptor is downwind of a source (see wind roses in Figure S3). The values in the south-east and north-west corners of the study domain are particularly low, so potential sources there would, on average, have low probability of being sampled at Ironbark.

510    The backward distribution for Burncluith (Figure 8b) is very similar, but since it is located north of Ironbark it would sample potential sources in the north-east better.

The two monitoring sites combined sample most part of the CSG sources in the domain (which was the prime objective of our monitoring).

**5.3 Bayesian inversion setup**

515 Assuming that emission rates are time invariant, we use all hourly methane data ($N_m$) from the two monitoring stations together in one combined Bayesian calculation to determine the total emission rates from gridded sources using Eq. (4). Since each hour corresponds to a unique meteorological condition, the use of all hours simultaneously provides the meteorological variability needed to achieve a better "triangulation" for source estimation. The greater the number of useful measurement hours, the greater the variability, and hence the better the constraining of the source. 
[revised manuscript text omitted]
 Gaussian prior with the individual bottom-up inventory emissions as its mean values has been used. The inversion retrieves the bottom-up emissions very well with a little scatter in the data points. The spatial distribution of the inferred emissions is presented in Figure 9c for this case, which is very similar to that of the inventory emissions in Figure 9a. As the prior uncertainty is increased to $\sigma_p = 10\%$ of the mean (Figure 11b), the uncertainty in the retrieved emissions gets larger, with a slight decrease in the correlation.

585 The total infrared emissions corresponding to Figure 11a and Figure 11a are $164.8 \times 10^6$ and $156.9 \times 10^6$ kg yr$^{-1}$, respectively – values somewhat smaller than the inventory total $173 \times 10^6$ kg yr$^{-1}$.

A comparison of Figure 9ca with the bottom-up inventory (Figure 9ab) indicates that some regions in the south-east, for example the strong coal mining source at the grid location (11, 4), and north west corners are not replicated as well bythat the inverse model is able to simulate the large emission rate in the region located just north of the Ironbark site. There is a strong 590 inventory emission on the eastern domain boundary which the model does not replicate. This is despite a perfect/strong prior with a relatively small uncertainty, and could be due to the fact that A possible reason for this is that the two monitoring locationsstations do not sample this source area sufficiently (see Figure 8). Extra monitoring stations and/or separate, narrower priors for sources that make very small contributions to methane at the two sites would be needed to cover these areas betterreduce the differences between Figure 9a and Figure 9b.

595 The above synthetic case results suggest that with only two monitoring locations the bottom-up inventory Gaussian prior works well and is, indeed, needed. Obviously, a small prior uncertainty biases the inferred emission distribution towards the prior $p(\mathbf{q})$, and what uncertainty level is selected depends on the available information supplied to the inversion. The synthetic case reveals that $\sigma_p \sim 5\%$ of the mean is needed to retrieve the bottom-up emissions. Thus, for a real inversion using the methane measurements one may expect that a tighter prior uncertainty would be needed. Further guidance on $\sigma_p$ can also comes from

600   comparison of the forward modelled methane concentrations using the inferred emissions with the methane observations from the two sites.

The synthetic case results also demonstratedsuggest that the regional inverse model formulated wasis stable and, feasible with MCMC., and credible as evident from its getting the total emissions nearly right and replicating the largest emission area reasonably well with only a broad prior and two monitoring locations, but at the same time requiring a relatively small prior

605   uncertainty. The synthetic case considered is an overly demanding case because the prior used is not very informative, compared to the real inversion cases considered in the next section in which the bottom-up inventory emissions allow the option of a better prior.

[Figure]

610   **Figure 9.** (a) **Emission rates of CH$_4$ (kg yr$^{-1}$ gridcell$^{-1}$) (a) based on the bottom-up inventory, (b)** estimated by the synthetic inversion **using a uniform Gaussian prior with an uncertainty of $\sigma_p$ = 5% of the mean for each source,** and (b) bottom-up inventory emission rates**, and (c) estimated by the synthetic inversion using the bottom-up inventory in (b) as a Gaussian prior with an uncertainty of $\sigma_p$ = 5% of the mean for each source. There are 11 × 11 sources.**

[Figure]

615

**Figure 10.** Scatter plot of the bottom-up inventory methane emission rates (g s⁻¹ per source) versus those inferred from the inverse (top-down) methodology for the synthetic case involving a uniform Gaussian prior with a prior uncertainty of (a) $\sigma_p$ = 5% and (b) $\sigma_p$ = 10% of the mean for each source. The number of sources is 11 × 11. The dash-dot line is the mean value of the prior, the dashed line is the 1:1 line (i.e. perfect agreement) and the solid line is the least-squares fit.

620

[Figure]

**Figure 11.** Scatter plot of the bottom-up inventory methane emission rates (g s⁻¹ per source) versus those inferred from the inverse (top-down) methodology for the synthetic case involving the inventory source emissions as the mean of a Gaussian prior with a prior uncertainty of (a) $\sigma_p$ = 5% and (b) $\sigma_p$ = 10% of the mean for each source. The number of sources is 11 × 11. The dashed line is the
625 1:1 line (i.e. perfect agreement) and the solid line is the least-squares fit.

**7 Inversion using the methane measurements**

We now use the filtered methane measurements from the two monitoring stations to quantify emissions using our inverse methodology. The above synthetic case results have revealed that a good, tight prior is needed to infer emissions within the selected domain using concentrations from the two monitoring locations. One may, of course, ask as to how the source
630    inference using the real-world measurements is influenced depending on the type of prior that may be available, ranging from a non-informative one to the most informative we have, i.e. the bottom-up inventory.

We use tThe same filtered methane observations as used in the forward transport modelling (so $N_m$ = 10581) are used in a̶one single Bayesian inverse run. , with tThe uncertainty in the measurements is $\sigma$ = 3.5 ppb based on previous calculation and the modelled uncertainty is $\sigma_m$ = 20% of the mean̶modelled concentration, as used in the synthetic inversion(
[revised manuscript text omitted]
 also influenced by grazing cattle, but the inferred emission seasonality for the CSG area cannot be linked with grazing cattle seasonality.

Another potential contributor to the temporal variability in the inferred emissions in Figure 17 is the seasonality of the winds in the area which influence the source-receptor relationships. We have not explored this possibility here.

[revised manuscript text omitted]
, whereas the increase is the smallest at 3.9% in the CSG subdomain, and is largest at 8.5% in the non-CSG region. Overall, this increase is expected because the increase in the concentration signal by 2.8 ppb as a result of the use of the alternate background (which is 2.8 ppb lower than the original background) needs to be to accounted for by the inversion by increasing the amount of inferred emissions. We also find that the amount of increase in the inferred emissions with the alternate background is almost uniformly spread through the study domain, and there are no significant spatial distributional shifts in the inferred emissions with the two background choices. This means that if these emissions are used in a forward model simulation, they would lift the modelled concentrations throughout the region by a very similar amount (probably by 2.8 ppb).

[revised manuscript text omitted]
 wetlands and thus methane emissions from grazing, influenced by rainfall. This correlation for the 3-monthly inferred emissions forfrom the full domain (Figure 17Figure 17a) is 0.71 and it is -0.06 for those from the CSG subdomain (Figure 17Figure 17b). It is reasonable to assume thatAssuming that the higher the rainfall the higher the grazing cattle (and wetland) emissions, and in that case these $r$ values indicate that the seasonal variability inof the inferred emissions within the full domain is, to a lesser degree, also
810     influenced by such emissionsgrazing cattle. However,, but the inferred emission seasonality withinfor the CSG area does not correlate with rainfall,cannot be linked with grazing cattle seasonality meaning that the emission seasonality is possibly dominated by the CSG sources.

        Another potential contributor to the temporal variability in the inferred emissions in Figure 17Figure 17 is the seasonality of the winds in the area which influences the source-receptor relationships. We have not explored this possibility here.

[revised manuscript text omitted]

There are possibly other and better ways of calculating the background methane concentration, such as having methane measurements at many locations around the perimeter of the study domain (which is often subject to operational and budget constraints) or modelling methane at much larger scale, preferably global, with data assimilation, which could then provide concentration boundary conditions needed for the regional modelling.

895 **Table 2: Inferred emissions (×10$^6$ kg yr$^{-1}$) obtained using: the methane background averaged over the two sites (as used in the paper, Case 3c), the individual methane background from the two sites, and the alternate methane background calculated using the Cape Grim baseline methane data (see Supplement S5). The values in the parentheses are % change over the inferred emissions using the averaged background. The bottom-up inventory emissions are also included for comparison.**

| Methane background | Total | CSG subdomain | Non-CSG subdomain |
|---|---|---|---|
| Average background (as used in this paper) | 165.8 | 63.6 | 102.2 |
| Separate backgrounds from the two sites | 164.8 (-0.6%) | 62.7 (-1.4%) | 102.1 (-0.1%) |
| Alternate background (see Supplement S5) | 177.0 (+6.8%) | 66.1 (+3.9%) | 110.9 (+8.5%) |
| Bottom-up inventory emissions | 173.2 (+4.5%) | 47.7 (-25%) | 125.5 (+22.8) |

900

**8 Conclusions**

This paper presented quantification of methane emissions from the CSG producing Surat Basin, an area of $350 \times 350$ km$^2$ in Queensland, Australia. The 2015 bottom-up methane emission inventory served as a very useful prior in our regional inverse methodology based on a Bayesian inference approach that utilised hourly-mean CH$_4$ concentrations

905 monitored at the Ironbark and Burncluith stations for 1.5 years, hourly source-receptor relationship, and an MCMC technique for posterior PDF sampling.

The largest contribution to the emissions in the bottom-up methane inventory was from grazing cattle (~50%), cattle feedlots (~25%), and CSG processing (~8%), with the aggregate emissions in the study area being approximately $173.2 \times 10^6$ kg CH$_4$ yr$^{-1}$. Although the forward transport modelling with the bottom-up emissions yielded a credible simulation of the suitably

910 filtered observed methane concentrations, about 15% of the higher-end concentration observations were underestimated.

 The importance of specifying a suitable prior in the Bayesian inference was made apparent by the synthetic

915 inversion, demonstrating  the use of the bottom-up inventory with a narrow uncertainty as being a good choice for that purpose when only two monitoring locations available. For inversion with the real methane measurements, a Gaussian prior having mean values taken the same as the bottom-up emissions with an uncertainty

equal to 3% of the mean yielded the best emission distribution, as evident from its performance in faithfully reproducing the measured methane concentration time series. This inverse setup yielded a domain-wide emission of $165.8 \times 10^6$ kg $CH_4$ yr$^{-1}$

920 which is very slightly less than the one obtained from the bottom-up inventory. However, within a subdomain covering all the CSG source areas, the inferred emission $63.6 \times 10^6$ kg $CH_4$ yr$^{-1}$is 33% larger than that deduced from the bottom-up inventory. The dominant localised inventory emissions in this area are from CSG, followed by feedlots. Since feedlots are scattered throughout the domain including the non-CSG areas from where there is no indicationinference of higher emissions, it is plausible that the increase in the inferred emissions would mainly correspond to CSG as the source sector.

925 Despite the amount of concentration data going into the seasonal inversion being relatively limited, the We also inferred seasonal variation of methane emissions fromwithin the non-CSGfull study domain, and CSG and 
[revised manuscript text omitted]